# Direct observation of photoinduced carrier blocking in mixed-dimensional 2D/3D perovskites and the origin

Dejian Yu[1,2], Fei Cao[1,2], Jinfeng Liao[2], Bingzhe Wang[2], Chenliang Su [1] ✉ & Guichuan Xing [2] ✉

Mixed-dimensional 2D/3D halide perovskite solar cells promise high stability but practically deliver poor power conversion efficiency, and the 2D HP component has been held as the culprit because its intrinsic downsides (ill charge conductivity, wider bandgap, and strong exciton binding) were intuitively deemed to hinder carrier transport. Herein, we show that the 2D HP fragments, in fact, allow free migration of carriers in darkness but only block the carrier transport under illumination. While surely limiting the photovoltaic performance, such photoinduced carrier blocking effect is unexplainable by the traditional understanding above but is found to stem from the trap-filling-enhanced built-in potential of the 2D/3D HP interface. By parsing the depth-profile nanoscopic phase arrangement of the mixed-dimensional 2D/3D HP film for solar cells and revealing a photoinduced potential barrier up to several hundred meV, we further elucidate how the photoinduced carrier blocking mechanism jeopardizes the short-circuit current and fill factor.

Halide perovskites (HPs) have attracted considerable attention for solar cells in recent years due to their superb light-matter interaction properties[1]. Although traditional 3D HPs have realized continuous breakthroughs in power conversion efficiency (PCE), they suffer from poor stability towards moisture, light, and heat[2]. Structure modification of HPs was actively attempted to mitigate the instability, and the so-called 2D HPs appeared credible[3–5]. 2D HPs can be viewed as partitioned 3D HPs by bulky ligands, and they could be subcategorized as $\langle 100 \rangle$, $\langle 110 \rangle$, and $\langle 111 \rangle$-oriented types depending on the partitioning manner[6]. The most widely studied Ruddlesden-Popper type (belonging to the $\langle 100 \rangle$-oriented type) respects a formula of $L_2A_{n-1}B_nX_{3n+1}$, where L is an organic ligand; A is methylammonium (MA), formamidine (FA), or Cs; B is Pb or Sn; X is a halogen Cl, Br, or I; $n$ denotes the HP layer number between two ligand layers. The atomically-dense hydrophobic ligand layer protects the inner HP units from moisture[4], and it also lowers the total Gibbs energy to improve the photothermal stability of 2D HPs[7].

The mixed-dimensional 2D/3D HP film promises to combine the superior optoelectronic properties of 3D HPs and the robustness of 2D HPs. Besides, the 2D HP component also serves as a defect passivator to suppress lossy recombination[8–12]. However, the practical PCE of mixed-dimensional 2D/3D HP solar cells is unsatisfactory, for which the 2D HP component is conventionally held accountable because its intrinsic downsides (ill charge conductivity, wider bandgap, and strong exciton binding) are believed to hinder carrier transport. Nevertheless, efforts centering on these aspects seem to make limited improvement. For the $n$ = 4-6 solar cells that better balance the performance and stability, the PCE barely exceeds 20% but usually falls within the range of 16–20%[13–23], which lags far behind that of the 3D HP counterparts (25.7%, certified by National Renewable Energy Laboratory, USA).

In this work, to test the conventional opinion, we design a proof-of-concept mixed-dimensional 2D/3D HP film prototype with the 3D domains sufficiently isolated by 2D HP linkers, which thus govern the global carrier migration. We find the 2D HP linkers allow free migration

[1]SZU-NUS Collaborative Innovation Center for Optoelectronic Science & Technology, International Collaborative Laboratory of 2D Materials for Optoelectronics Science and Technology of Ministry of Education, Institute of Microscale Optoelectronics, Shenzhen University, Shenzhen 518060, China. [2]Joint Key Laboratory of the Ministry of Education, Institute of Applied Physics and Materials Engineering, University of Macau, Taipa, Macao SAR 999078, China. ✉e-mail: chmsuc@szu.edu.cn; gcxing@um.edu.mo

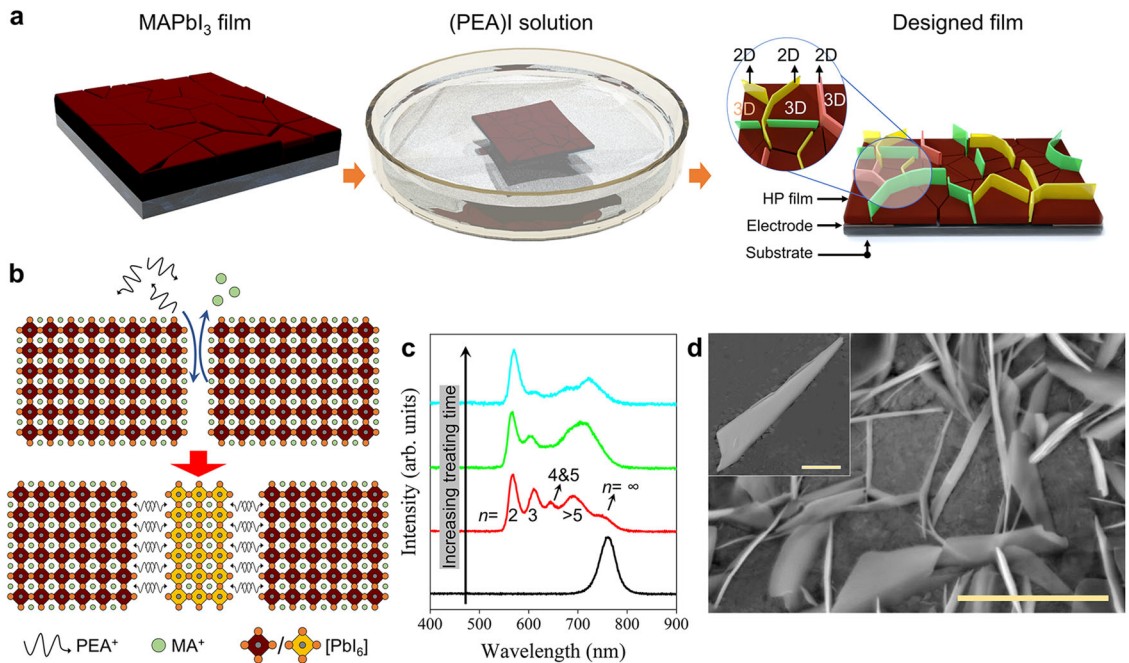

**Fig. 1 | Isolating 3D HP grains by 2D HP platelets. a** Schematic diagram of the experimental fabrication process of the GIHP film. **b** Schematic illustration of the cation exchange-induced growth of 2D HP at the boundaries of 3D HP grains. **c** Treating time-dependent PL spectra of the film. **d** The SEM micrograph of the obtained sample. The scale bar in is 5 µm. Inset: Zoomed view of the 2D HP platelet inserted in the grain boundary, and the scale bar is 500 nm.

of carriers in darkness but block the carrier transport under illumination, which is unexplainable by the conventional understanding above. The transient spectroscopy characterization unveils that such a photoinduced carrier blocking (PCB) effect stems from the trap-filling-enhanced built-in potential of the 2D/3D HP interface. We further parse the depth-profile phase arrangement of the mixed-dimensional 2D/3D HP film for solar cells and reveal a photoinduced potential barrier up to several hundred meV to carriers. Accordingly, how the PCB mechanism blunts the short-circuit current ($J_{sc}$) and fill factor (FF) is elucidated. These results point out that previous efforts revolving around the 2D HP component might be insufficient or even off-target. The future focus for improving mixed-dimensional 2D/3D HP solar cells should be redirected to the photoinduced interface effects between the 2D and 3D HP components.

## Results

### Design of the 3D HP grain-isolated film prototype

The impact of the 2D HP component on carrier transport is difficult to study in traditional mixed-dimensional 2D/3D HP films because there are complex parallel paths for carrier migration[24,25]. Therefore, the negative impact of the 2D HP fragments, if at all, would be concealed in ensemble measurements. We envision that if the 2D HP linker sufficiently cuts the connection between 3D HP grains to become the only route for inter-3D-grain carrier transfer, its impact would expose. Such a proof-of-concept prototype could be more easily realized in a lateral configuration. Experimentally, a 3D MAPbI₃ film fabricated by spin coating was treated with a solution containing phenylethylamine (PEA⁺), as illustrated in Fig. 1a. A pivotal step is to slacken the grain boundaries a bit (by using toluene as the antisolvent) so that the PEA⁺ could penetrate easily. Subsequently, a spontaneous MA⁺-to-PEA⁺ exchange would occur to induce a 3D-to-2D dimensionality transition (Fig. 1b)[26,27]. Without engineering the grain boundaries, such transition would happen on the surface[28], or randomly (Supplementary Fig. 1).

The transition course was recorded by the treating time-dependent scanning electron microscope (SEM) characterization (Supplementary Fig. 2): In the grain boundaries, platelet-shaped crystals that accord with the structure of 2D HPs emerge and gradually

grow over time. In line with the growth of the platelet crystals, the one-dimensional X-ray diffraction (XRD) characterization shows the nucleation and progressive strengthening of the 2D HP peaks (Supplementary Fig. 3). Consistently, the photoluminescence (PL) spectra (Fig. 1c) reveal the appearance of the characteristic peaks of 2D HPs and, concurrently, a waning relative peak intensity of the 3D HP, which confirms the 3D-to-2D HP transition. A slight blueshift of the 3D HP PL peak that agrees with the grain size decrease (caused by the edge consumption) can be observed after the treatment. Therefore, we can obtain an as-designed HP film with the connection between 3D HP grains sufficiently, if not entirely, cut by 2D HP platelets in the grain boundaries (Fig. 1d). The prolonged PL lifetime after the treatment (Supplementary Fig. 4 and Supplementary Note 1) suggests a passivation effect by the 2D HP linker, and the temperature-dependent conductivity measurement reveals a shallower trap distribution in it than in the 3D HP film (the carrier activation energies are 22 meV & 8 meV for the former vs. 23 meV & 15 meV for the latter, Supplementary Fig. 5 and Supplementary Note 2). For simplicity, this sample is termed GIHP (grain-isolated HP) film henceforth.

### Blocked carrier transport under illumination

The above GIHP film was deposited on a channeled ITO substrate to form a two-terminal photoresistor for measurement (Fig. 2a). According to the ultraviolet photoelectron spectroscopy (UPS) characterization (Supplementary Figs. 6 and 7 and Supplementary Note 3), the 2D and 3D HP form a type-I band alignment with the 2D HP side being weakly p-type and the 3D HP side being weakly n-type, and the conduction band minimum (CBM) offset is larger than the valence band maximum (VBM) offset. The VBM positions of the 2D and 3D HPs are in the range of −6.24 eV ~ −6.35 eV versus the vacuum level, and the Fermi level of ITO is −4.75 eV ~ −4.60 eV. Therefore, the energy alignment of the GIHP/ITO interface allows easy inflow/outflow of electrons but sets an overlarge potential barrier for holes so that we can obviate the interference caused by bipolar carrier transport. Indeed, the I-V curve acquired in darkness presents a clear Ohmic contact property (Fig. 2b). It also means no rectification characteristic of the 2D/3D HP junction, and electrons can freely pass through the 2D HP linker

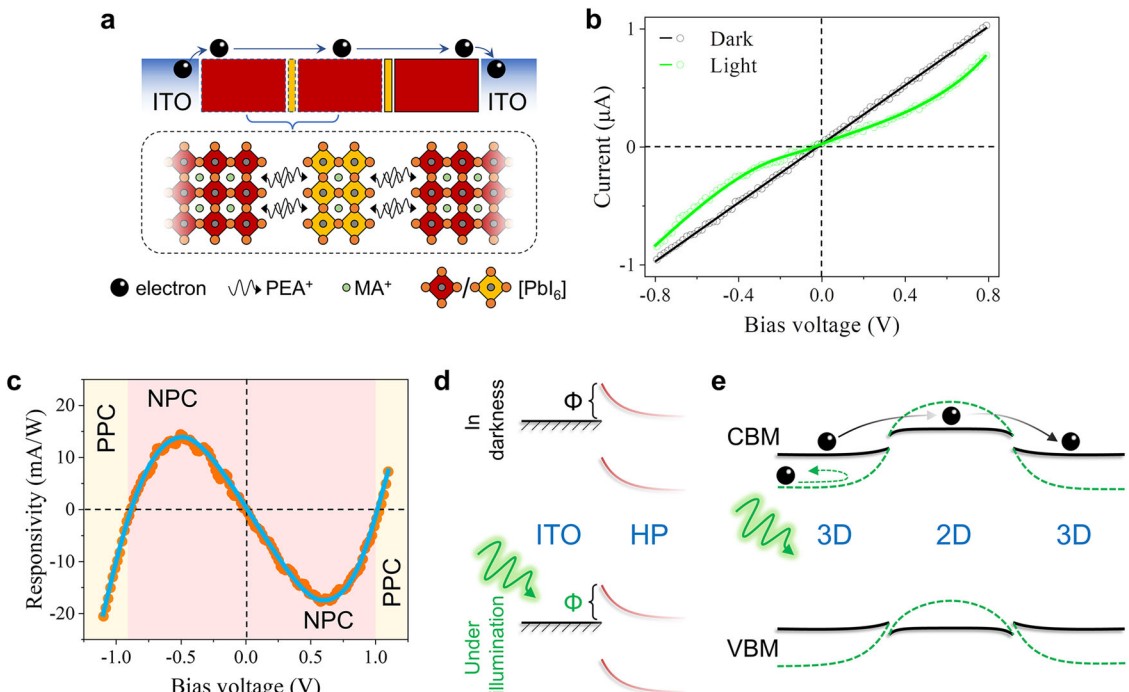

**Fig. 2 | Carrier transport of the GIHP film in darkness and under illumination. a** Schematic diagram of the two-terminal ITO/GIHP film/ITO device configuration for optoelectronic measurements. **b** The corresponding *I-V* characteristics in darkness and under illumination. **c** The bias-dependent responsivity of the GIHP film. **d** Schematic diagrams of the unvaried Schottky barrier height of the ITO/GIHP junction during the switch between darkness and illumination. **e** Schematic of the light-induced energy landscape variation within the GIHP film. The black solid lines represent the energy band diagram in darkness, and the green dashed lines represent the energy band diagram under illumination.

despite the CBM offset between the 2D and 3D HPs. We can also infer free transport of holes in the GIHP film given the smaller VBM disparity.

However, under photoexcitation, the GIHP film presents abnormal negative photoconductivity (NPC), i.e., the light current is smaller than the dark current. The *I-t* curve measured under a periodic light on/off test (Supplementary Fig. 8, @ 532 nm and 460 nm excitation wavelengths, 0.5 V) confirms stable NPC. This phenomenon is unexplainable by the intrinsic setbacks of 2D HPs, namely the ill charge transport, wider bandgap, and strong exciton binding, as they have no reason to suppress the photoconduction effect. By scrutinizing the *I-V* characteristic, it is not difficult to find that such NPC is caused by an Ohmic-to-Schottky contact property transition, which signifies a light-induced potential barrier (absent in darkness but present under illumination) to electrons at some interface. Indeed, as the illumination intensity increases, the photocurrent (defined as light current minus dark current, the value is negative) under a bias voltage of 0.5 V decreases (Supplementary Fig. 9), corroborating it is a light-triggered potential barrier. The potential barrier model is also supported by the voltage-switchable photoconductivity polarity of the GIHP film, i.e., NPC in the low-voltage realm and positive photoconductivity (PPC) in the high-voltage realm (Fig. 2c). This means if photogenerated electrons are empowered with high kinetic energy, they can surmount the light-induced potential barrier to freely migrate again. Note that the mechanism for the NPC here should differ from the traditional NPC caused by molecule adsorption[29,30], photothermal effect[31], or photobolometric effect[32,33], because their impact is to reduce the carrier mobility rather than to alter the interface contact property. Given that the Schottky barrier (Φ) height of the ITO/GIHP junction (essentially a metal-semiconductor junction) would not vary upon photoexcitation (Fig. 2d), it is reasonable to infer that the light-induced barrier stems from the 2D/3D HP interface.

We can thus reason the light-induced energy landscape variation of the 2D/3D interface as schematically illustrated in Fig. 2e: In

darkness, despite the presence of CBM (VBM) offset, electrons (holes) can freely transport through the 2D linker to realize inter-3D-grain transfer; When the GIHP film is photoexcited, the built-in potential of the 2D/3D HP interface increases, hence setting a potential barrier to repel electrons (holes).

To understand the carrier dynamics behind the light-enhanced built-in potential of the 2D/3D HP interface, we then carried out the transient absorption (TA) spectra characterization (Supplementary Fig. 10). As shown in Fig. 3a, both the 2D (*n* = 1–4) and 3D HP phases present consistent three-stage TA kinetics, signaling highly correlated carrier dynamics between them. The first stage (labeled as ①) reveals rapid photobleaching (PB) decay for the 2D HPs but a PB rise for the 3D HP, which suggests a 2D-to-3D charge- or energy transfer. The corresponding time-resolved PL spectrum (Fig. 3b) shows an initial PL intensity rise for the 3D HP but a decay for the 2D ones, and the feature time closely coincides with that of ①. Therefore, ① should be (mainly) ascribed to the well-known Förster energy transfer from wide-bandgap 2D phases to the narrow-bandgap 3D phase[34,35].

For the second stage (labeled as ②), all the phases maintain a plateau of the PB signals, indicating suppressed carrier recombination, which could plausibly be assigned to a carrier separation process driven by the built-in potential according to the energy landscape of the 2D/3D interface (Fig. 2e). The carrier separation nature of ② can be confirmed in the following way: Different from the energy transfer in ① that accelerates the PB decay of the 2D HPs (as the energy donor), the carrier separation process in ② should inject holes (from the n-type 3D phase) to the p-type 2D HPs to slow their PB decay. Therefore, a stronger carrier separation (of ②) would dwarf the energy transfer (of ①) to shorten the feature time of ① and vice versa. Considering that the carrier separation efficiency is decisively impacted by the built-in potential of the 2D/3D HP interface, we then explore the interrelationship between the feature time of ① and the built-in potential in a series of 2D/3D HP interfaces (butylamine-type, octylamine-type, and

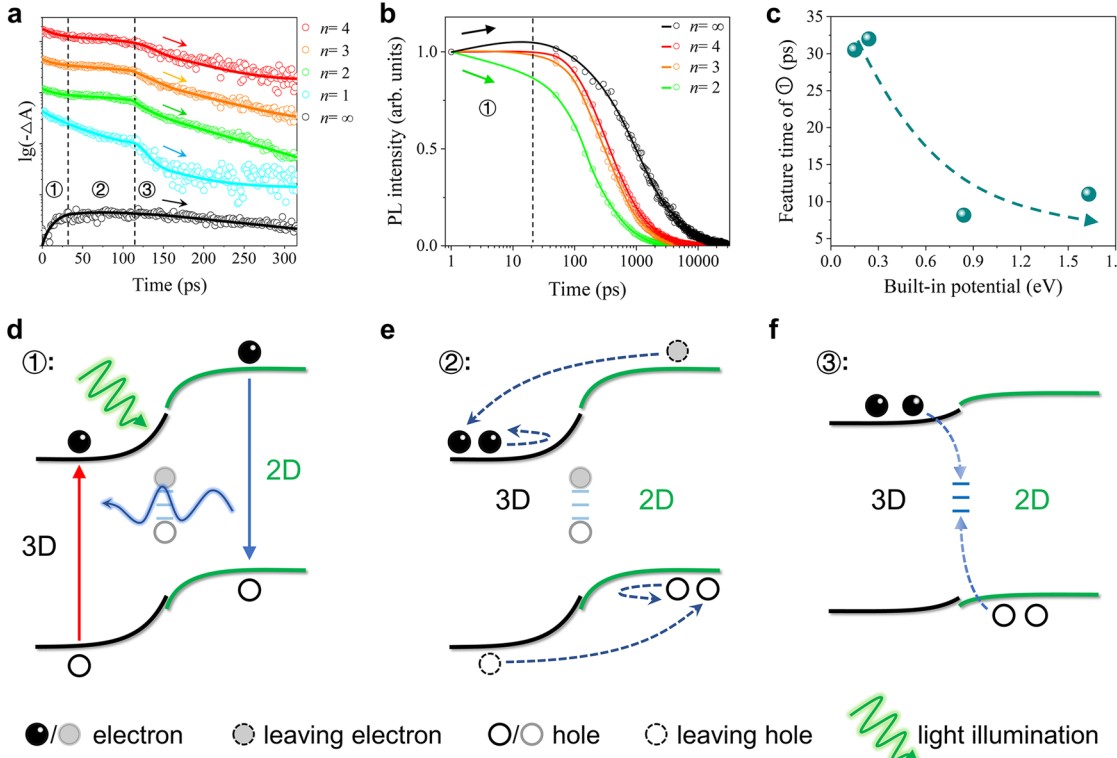

**Fig. 3 | Carrier dynamics of the 2D/3D HP interface. a** Staged TA kinetics probed at the characteristic wavelengths of dissimilar phases (520 nm for $n = 1$, 569 nm for $n = 2$, 614 nm for $n = 3$, 644 nm for $n = 4$, 760 nm for $n = \infty$). **b** Transient PL kinetics of the $n = 2$-4 2D phases and the $n = \infty$ 3D phase. **c** The dependence of the feature time of ① on the built-in potential of the 2D/3D HP interface. **d**–**f** Schematic illustrations of the carrier dynamics of ①, ②, and ③.

phenylpropamine-type in addition to the PEA-type one. Supplementary Figs. 11–14 and Supplementary Note 4). These 2D/3D HP interfaces present the same three-stage TA kinetics (Supplementary Fig. 15) as the PEA-type so that the extracted parameters are self-consistent. As shown in Fig. 3c, the feature time of ① generally decreases as the built-in potential increases, which supports the carrier separation nature of ②. By the way, ① therefore should be a superposition of the energy transfer process and the carrier separation process due to its very same energy landscape as ②, but the energy transfer process dominates in ①.

In the third stage (labeled as ③), the PB signals of both the 2D and 3D phases simultaneously start to decay, which points to the annihilation of opposite charges at the 2D/3D interface. Two features can be identified for such an interfacial recombination process: (i) It dominates only when the carrier concentration is low; (ii) It features the highest recombination rate through the entire carrier lifespan for both 2D and 3D HPs. Accordingly, ③ is attributed to a trap-assisted recombination process: At a high carrier concentration, the interfacial traps are filled and their effect is concealed; Only at a low carrier density, the interfacial traps are no longer filled and start to discernibly promote the lossy carrier recombination. Importantly, such interfacial carrier recombination of ③ is a reverse process of the carrier separation in ②. The sequential occurrence of the two opposite processes implies an energy landscape variation of the 2D/3D HP interface from ② to ③, i.e., the high built-in potential that results in carrier separation in ② decreases in ③. Otherwise, the electrons and holes would still be kept separated and cannot encounter at the 2D/3D HP interface. Given that unneutralized defects would screen the built-in electric field, the energy landscape change of the 2D/3D HP interface is credited to the trap de-filling process in ③.

Based on the above analysis, we can picture the overall photodynamics of the 2D/3D HP interface as below: Upon pulsed excitation, the compact and neat 2D/3D HP interface gives rise to 2D-to-3D energy transfer (Fig. 3d); In parallel, the interfacial traps are filled by the

photogenerated carriers so that the previously screened built-in potential in darkness is strengthened, thus forcing carrier separation (Fig. 3e). This corresponds to the photoinduced potential barrier of the GIHP film that blocks the inter-3D-grain carrier transfer; As the carrier concentration decreases, the interfacial traps vacate such that the built-in potential is back to a screened state, and the opposite charges are no longer separated but rather encounter at interfacial traps to recombine (Fig. 3f). This corresponds to the free transport of carriers in the GIHP film in darkness.

## Depth-profile phase arrangement

For the solar cell application, mixed-dimensional 2D/3D HP films tend to form a gradient phase distribution in the vertical direction, i.e., small-n phases on the top and large-n phases on the bottom or reverse, and this has been believed to construct a gradient energy band to promote carrier harvesting[13–15,17,22,36–38]. Here we show that such a phase gradient is practically a macroscopic average result, while at the nanoscopic scale, there is still a disordered phase arrangement. Several previous reports also mentioned this point[39,40], but a detailed investigation is still needed, as such local disordered phase arrangements would entitle the detrimental PCB effect to appear in mixed-dimensional 2D/3D HP solar cells.

A PEA$_2$MA$_4$Pb$_5$I$_{16}$ ($n = 5$) was fabricated by the typical hot-casting method with modification[4,41]. The absorption spectrum (Supplementary Fig. 16a) confirms its 2D/3D HP mixed nature by showing the corresponding characteristic absorption peaks. The PL spectrum (Supplementary Fig. 16b) collected from the top surface (the HP side) mainly presents the signal of 3D HP, while the emissions of the 2D ones become quite conspicuous when probing at the bottom surface (the quartz side), which suggests a macroscopic decrease of dimensionality (3D-to-2D) from top to bottom as consistent with previous reports[13,17,36,38,42]. To explore the detailed depth-profile phase composition, we conducted the two-photon PL mapping measurement

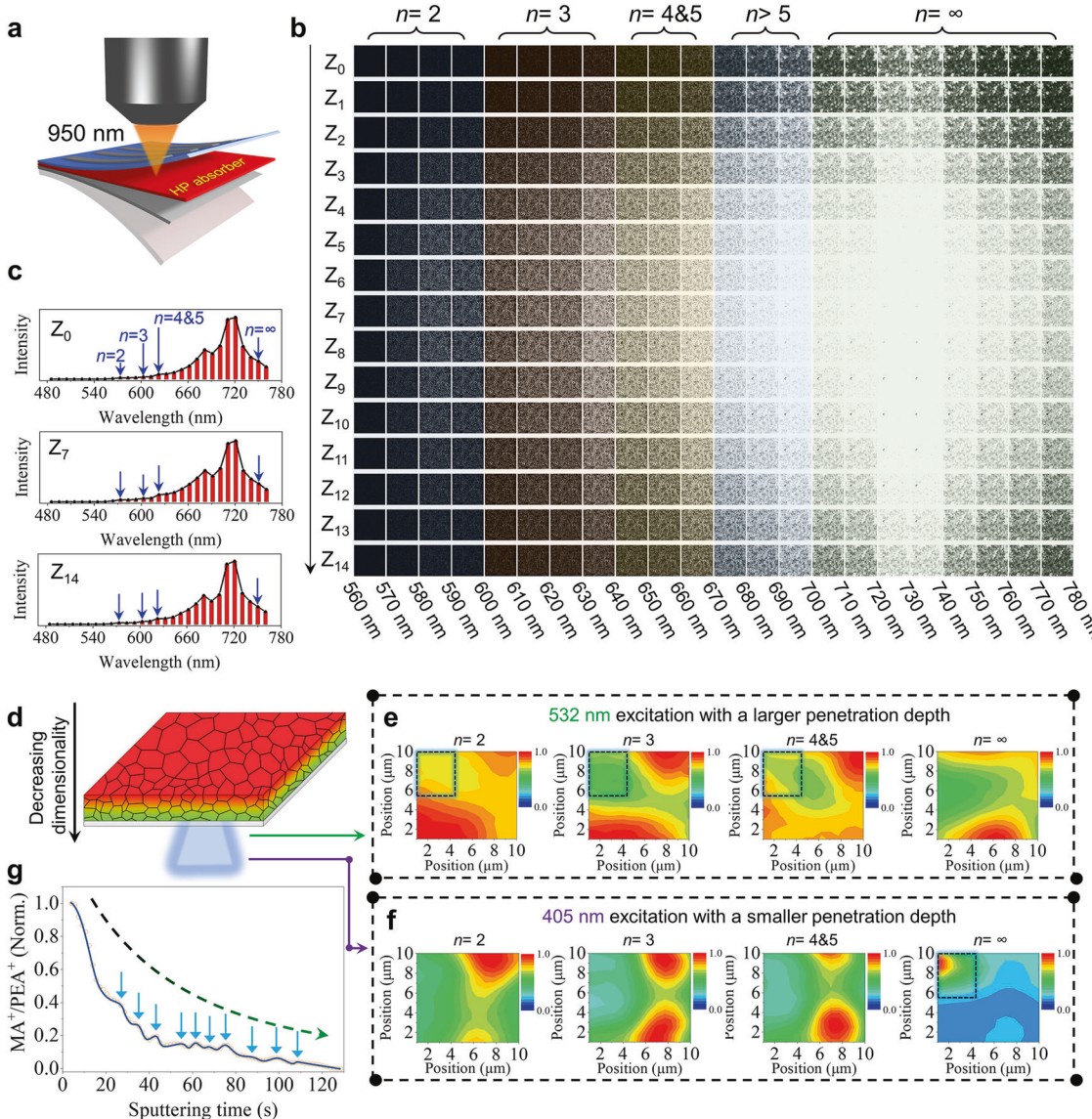

**Fig. 4 | Depth-profile phase arrangement in the mixed-dimensional 2D/3D HP film. a** Schematic diagram of the two-photon PL mapping. **b** The depth-profile two-photon PL mapping result. $Z_0$ and $Z_{14}$ represent the top and the bottom of the film, respectively. The other cross-sections are evenly spaced between $Z_0$ and $Z_{14}$ (denoted as $Z_1$, $Z_2$, $Z_3$, etc.). The PL mapping was conducted for each 10 nm band, and the relative brightness of the picture reflects the PL intensity. The monitoring area is 58.7 μm × 58.7 μm. The pumping wavelength is 950 nm. **c** Reorganized PL spectra of the $Z_0$, $Z_7$, and $Z_{14}$ sections. **d** Schematic diagram of the one-photon PL mapping probing at the bottom surface of the film. **e** One-photon PL mapping results of the $n = 2$, 3, 4 & 5 (the PL fingerprints are too close to differentiate), and ∞ phases by the 532 nm excitation wavelength. Corresponding tracing wavelength bands are 570–589 nm for the $n = 2$ phase, 613–634 nm for the $n = 3$ phase, 646–664.6 nm for the $n = 4$ phase, and 754–774 nm for the $n = ∞$ phase. **f** One-photon PL mapping results of the $n = 2$, 3, 4 & 5, and ∞ phases by the 405 nm excitation wavelength. **g** The variation of the $n$ value with depth inferred from the depth-profile $MA^+/PEA^+$ ratio. The blue arrows mark local regions of phase misarrangement.

(@950 nm) and found that each cross-section comprises mixed 2D and 3D HP phases. The two-photon PL measurement precisely detects the confocal point (Fig. 4a), and the PL imaging solely mirrors the situation of the cross-section scanned. The detecting area is 58.7 μm × 58.7 μm, the brightness reflects the overall PL intensity, and the measurement was implemented for every bandwidth of 10 nm from 560 nm to 780 nm. As shown in Fig. 4b, the 2D HP phases of $n = 2$, $n = 3$, and $n = 4$&5 (the PL fingerprints are too close to differentiate) are present in each cross-section in addition to the 3D HP phase ($n = ∞$). Their characteristic PL peaks can also be discerned in the PL spectrum obtained by reorganizing the results of each bandwidth (Fig. 4c). Note that the relative PL intensity change between 2D and 3D HPs cannot quantitively reflect their content variation due to different degrees of self-absorption to their emissions, and the surface traps indicated by the

dimmer images (weaker PL) near the top and the bottom surfaces could pose different influences to 2D and 3D HPs.

Further, to inspect the axial phase arrangement, we conducted the one-photon PL mapping at the backside of the film (Fig. 4d) and employed two excitation wavelengths of 532 nm and 405 nm. The 532 nm photon could penetrate through 34.2% of the film thickness (Supplementary Fig. 17 and Supplementary Note 5), which is about 3 times that of the 405 nm photon (11.7%). By selectively probing the characteristic PL bands, we can map the lateral distribution of the $n = 2$, $n = 3$, $n = 4$ & 5 2D phases, and the 3D phase. Contrasting the two-photon PL mapping, the phase distribution given by the one-photon PL mapping is an average result within the penetration depth of the excitation light. As shown in Fig. 4e, f, the phase distributions for all these phases significantly vary when switching the

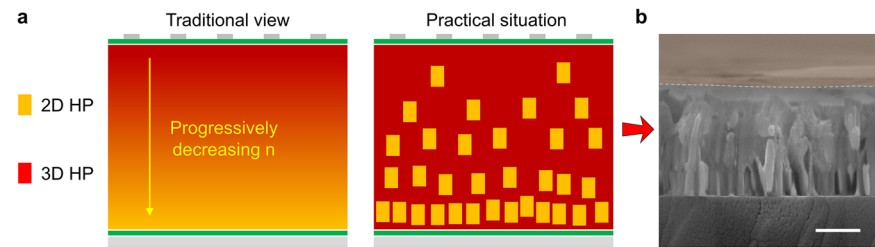

**Fig. 5 | Phase distribution in mixed-dimensional 2D/3D film for solar cells.**
**a** Schematic illustration of the depth-profile phase distribution of the gradient mixed-dimensional film in the traditional opinion (the left panel) and that in the practical situation (the right panel). The yellow rectangle (and the yellow part) represents the 2D HP component, and the red part represents the 3D HP component. **b** The cross-section SEM micrograph of the gradient mixed-dimensional film. The scale bar is 500 nm.

excitation wavelength, indicating an unmatched phase distribution between the deeper region (away from the bottom) and the shallower region (near the bottom). Such inconsistent axial phase distribution at different depths is certain to engender local phase misarrangement. For instance, within the shallow detection range (of the 405 nm excitation), the 3D phase is concentrated in the upper left corner of the probing area (Fig. 4f, the right panel, circled by a dashed rectangle), but there are $n$ = 2, 3, 4 & 5 2D HP phases above this corner (beyond the penetration depth of the 405 nm photon but within the penetration depth of the 532 nm photon. The left three panels of Fig. 4e, circled by a dashed rectangle). Such local phase misarrangement does not defer to the global 3D-to-2D transition from top to down, and it means carriers from above have to pass through the 2D HP linker before reaching the circled 3D HP grain below.

To generalize the conclusion of such nanoscopic phase misarrangement across the depth, we implemented the time-of-flight secondary ion mass spectrometer (TOF-SIMS) measurement to learn the precise depth-profile $n$ variation (Supplementary Fig. 18). The $n$ value variation can be acquired by monitoring the $MA^+$/$PEA^+$ ratio variation with depth according to the chemical formula of quasi-2D HPs. As shown in Fig. 4g, despite a global $n$ value reduction with depth as expected, conspicuous humps that correspond to a low-n/high-n/low-n phase arrangement can be observed.

Based on the above results, we can obtain the practical phase distribution of the so-called gradient mixed-dimensional HP film, as shown in Fig. 5a. Instead of following a progressive dimensionality transition, the film is a mixture of 2D and 3D phases for each axial cross-section. The 2D component accounts for a lower ratio near the top surface but a higher ratio near the bottom surface. Therefore, the 2D HP grains play an important role to interlink the 3D grains above and below them despite the parallel 3D HP paths. This model is consistent with the phase segregation model proposed by Lin et al.[39]. As shown in the cross-section SEM image of the film (Fig. 5b), discontinuous flake-shaped grains corresponding to 2D HPs can be clearly observed along the axial direction, which accords with the proposed phase distribution.

## In-plane PCB and its effect on solar cells
From the perspective of space, the PCB effect is confirmed along the out-of-plane direction of 2D HPs in the above GIHP film (the "plane" here refers to the (h00) facets of 2D HPs[43–45]). But due to a vertical orientation of the 2D component in typical mixed-dimensional 2D/3D HP solar cells (Supplementary Fig. 19), carriers are expected to transport along the in-plane direction. However, from the perspective of energy, the PCB effect would naturally occur in the in-plane direction similar to in the out-of-plane direction. As schematically illustrated in Fig. 6a–c: When the energy band of the 2D HP is upshifted relative to the 3D HP along the out-of-plane direction, the same band upshift would also occur along the in-plane direction.

To experimentally confirm the in-plane PCB effect, a vertical configuration of ITO/HP/Cu (Fig. 6d) was adopted for measurement. Cu has a close work function to ITO (4.7 eV for Cu and 4.6-4.75 eV for ITO) so that this configuration is almost energetically symmetric similar to the lateral ITO/GIHP film/ITO above, and it is also an electron-only configuration that excludes the interference from bipolar transport. Differently, in this vertical configuration, the NPC phenomenon is absent and thus could not serve to prove the in-plane PCB effect: On the one hand, the background dark current is so high (~mA level) that the photocurrent is largely masked; On the other hand, the 2D/3D HP interface is not likely to be the only carrier path as in the GIHP film, therefore, its NPC effect would be averaged out by other parallel PPC carrier paths. Alternatively, we could monitor the negative current ($I_{neg}$) plus positive current ($I_{pos}$) versus the absolute voltage value ($V_{abs}$) to gain insights (e.g., $I_{neg}$ at −0.15 eV plus $I_{pos}$ at 0.15 V, the corresponding $V_{abs}$ is 0.15 V). Here, the negative current $I_{neg}$ corresponds to the top-to-bottom electron transport, and the positive current $I_{pos}$ corresponds to the bottom-to-top electron transport. Therefore, the ($I_{neg}$+$I_{pos}$) versus $V_{abs}$ characteristic reflects the symmetry of electron transport. For perfect symmetry, ($I_{neg}$+$I_{pos}$) would be zero for any $V_{abs}$.

As presented above, the top side of mixed-dimensional 2D/3D HP films is rich in the 3D component and the bottom side is rich in the 2D component. In darkness, electrons would freely transport between 2D and 3D HPs to enable relatively symmetric electron transport (Fig. 6e); Under illumination, if the in-plane PCB effect exists, the photo-enhanced built-in potential of the 2D/3D HP interface would upshift the energy band of 2D HPs relative to the 3D HP, hence setting a global barrier to the top-to-bottom electron transport (Fig. 6f). Therefore, the ($I_{neg}$+$I_{pos}$) versus $V_{abs}$ trends measured in darkness and under illumination should be different.

Figure 6g, h reveal a general Ohmic contact property the same as the lateral ITO/GIHP film/ITO configuration. As shown in Fig. 6i, in darkness, the ($I_{neg}$+$I_{pos}$) value increases with $V_{abs}$ (more approaching zero). This means the ITO/bottom contact owns a slightly higher Schottky barrier than the top/Cu contact, therefore, the ITO-to-Cu (bottom-to-top) electron transport is less efficient than the Cu-to-ITO (top-to-bottom) electron transport, but the difference vanishes at high $V_{abs}$ because the Schottky barriers are easy to surmount. Such Schottky barrier disparity agrees with the higher 2D perovskite portion at the bottom of the mixed-dimensional film. In contrast, one-sun illumination engenders a reverse trend of ($I_{neg}$+$I_{pos}$) versus $V_{abs}$, i.e., the top-to-bottom electron transport (corresponds to $I_{neg}$) grows with $V_{abs}$ more than the bottom-to-top electron transport (corresponds to $I_{pos}$). Indeed, the trend presents a conspicuous diode-like feature that suggests a global photoinduced potential barrier to the top-to-bottom electron transport, which agrees with the effect of the in-plane PCB effect analyzed above. A repeated measurement (Supplementary Fig. 20) presents the same result, thus excluding experimental errors. However, as a comparison, the vertical ITO/MAPbI$_3$/Cu configuration is

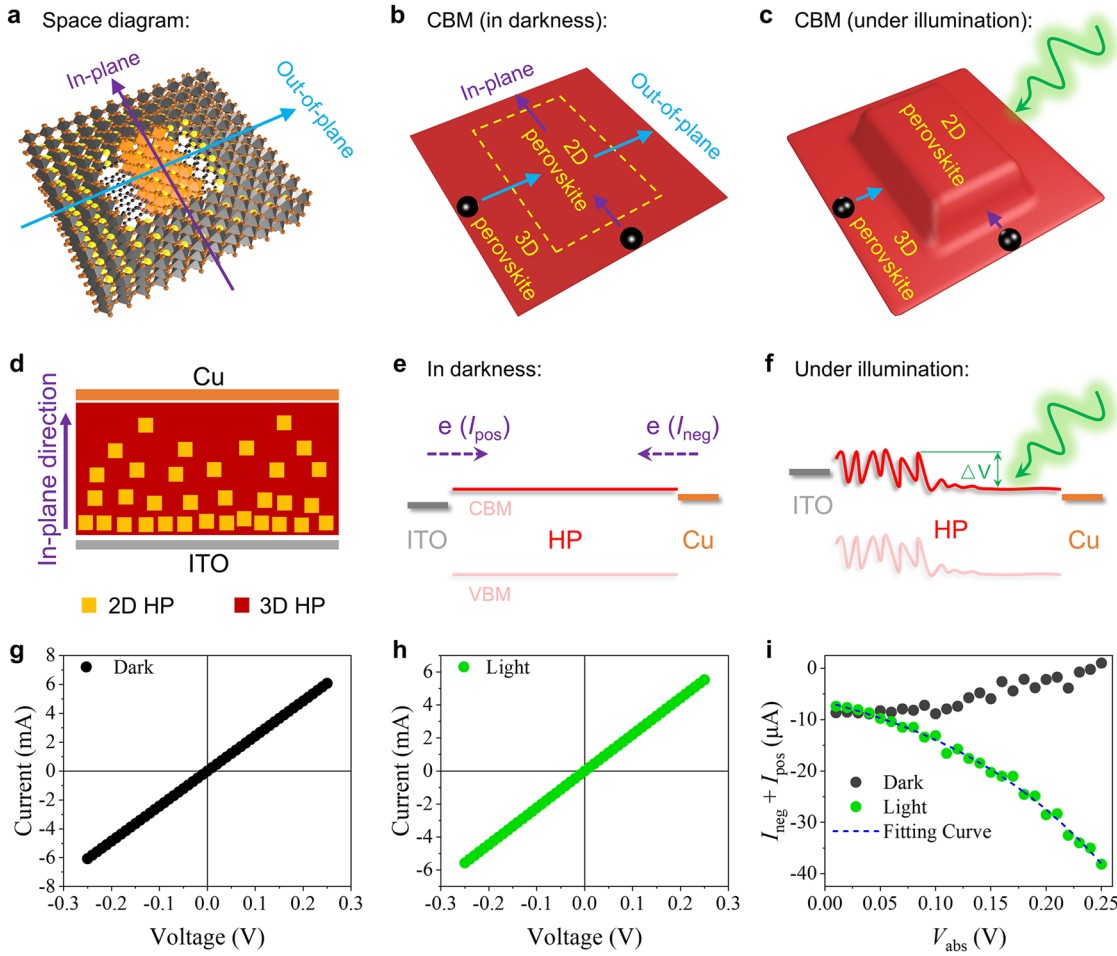

**Fig. 6 | Theoretical analysis and experimental verification of the PCB effect along the in-plane direction of 2D HPs.** a Space diagram of the 3D/2D/3D HP configuration in typical mixed-dimensional 2D/3D HP films for solar cells. Schematic illustration of the corresponding CBM **b** in darkness and **c** under illumination. **d** Schematic diagram of the ITO/HP/Cu configuration for the in-plane direction measurement; Schematic diagram of the corresponding energy landscapes (**e**) in darkness and **f** under illumination. e represents electrons. $\triangle V$ denotes the photoinduced global potential barrier; The *I-V* characteristics of the vertical ITO/PEA$_2$MA$_{2.67}$Pb$_{3.67}$I$_{12}$/Cu (**g**) in darkness and **h** under illumination. **i** The corresponding ($I_{neg}+I_{pos}$) versus $V_{abs}$ characteristics in darkness and under one-sun illumination.

free of such photoinduced change (Supplementary Fig. 21 and Supplementary Note 6).

The photoinduced global barrier could further be fitted (by a single exponential function) to be 165.5 meV. Such a high potential barrier far exceeds the thermal perturbation energy (~25 meV at room temperature). Furthermore, the universal presence of the in-plane PCB effect is confirmed in various mixed-dimensional films with different compositions, and the reliability of these results is also supported by repeated measurements (Supplementary Figs. 22–24). The corresponding photoinduced potential barriers in these films are summarized in Table 1 and Supplementary Table 1. The highest photoinduced barrier reaches even up to 242.0 meV.

Overall, the impact of the PCB effect on the carrier harvesting of mixed-dimensional 2D/3D HP solar cells is illustrated in Fig. 7a, b: Although freely moving in darkness, electrons would be greatly hampered under illumination, and holes would be also deeply trapped. By comparing the top mixed-dimensional 2D/3D HP solar cells (MA-based

or FA-based but not Cs-based, $n = 4$-6) to the best-performing 3D HP counterparts in literature, it is not difficult to find that $J_{sc}$ and FF rather than the open-circuit voltage ($V_{oc}$) limit their PCE (Fig. 7c, d). This agrees well with the negative impact of the PCB effect: In the short-circuit condition, photogenerated carriers are hampered by the light-induced potential barrier, especially in the quasi-neutral region, so that the $J_{sc}$ is reduced. Note that the inferior $J_{sc}$ also partly originates from the wider bandgap of mixed-dimensional HPs; Upon a certain bias voltage, carriers have to first surmount the light-induced barrier before could freely move such that the $J_{sc}$ is lower than it otherwise could be without the PCB effect, thus causing a lower FF. Noteworthily, the PCB mechanism also justifies the usually better optoelectronic performance of the 2D-on-3D bilayer configuration[46–49]. Because the 2D-on-3D bilayer configuration is essentially a unidirectional 2D/3D HP heterojunction on the surface, the photo-enhanced built-in potential of the heterojunction only helps to separate the electrons and holes, thus improving the charge collection efficiency of the solar cell rather than causing a detrimental PCB effect (Supplementary Fig. 25 and Supplementary Note 7).

Accordingly, we reason that the PCE of mixed-dimensional 2D/3D HP solar cells could be further improved by increasing the $J_{sc}$ and FF by obviating the PCB effect, as illustrated in Fig. 7e. The built-in potential stems from the inconsistent (self-) doping levels of the 2D and 3D HP components. Therefore, delicate doping across the

**Table 1 | Summary of the photoinduced potential barriers in various mixed-dimensional 2D/3D HP systems**

| A-site cation | MA | Cs$_{0.5}$MA$_{0.5}$ | FA$_{0.5}$MA$_{0.5}$ | MA$_{0.5}$Cs$_{0.25}$FA$_{0.25}$ |
|---|---|---|---|---|
| Photoinduced barrier (meV) | 165.5 ($n = 3.67$) | 204.5 ($n = 3.67$) | 126.8 ($n = 3.67$) | 242.0 ($n = 4$) |

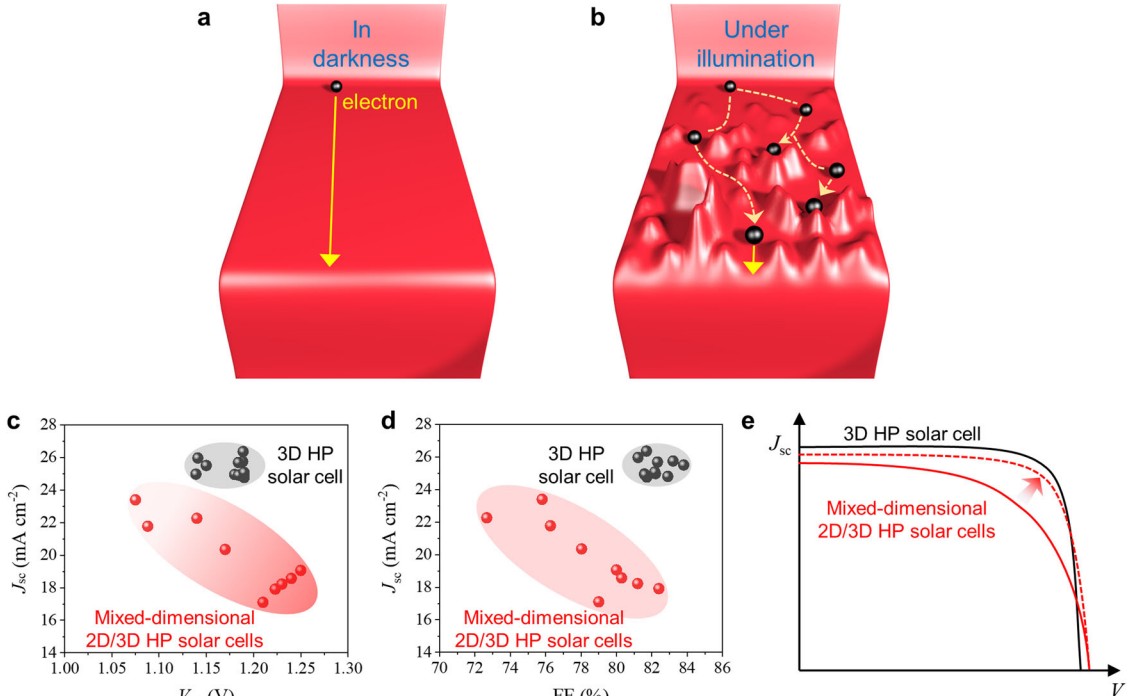

**Fig. 7 | How the PCB effect limits the solar cell performance.** Schematic illustration of the carrier transport within mixed-dimensional 2D/3D HP films **a** in darkness and **b** under illumination. **c**, **d** The comparison of $V_{oc}$, FF, and $J_{sc}$ between the top mixed-dimensional solar cells[13–19, 21, 23], and the best-performing 3D HP solar cells in literature[52–62]. **e** Illustrative J-V characteristics of the 3D HP solar cells (black line) and the mixed-dimensional HP solar cells. The red solid line represents the state-of-the-art J-V characteristic of mixed-dimensional 2D/3D HP solar cells, and the red dashed line represents the expected J-V characteristic of them without the PCB effect.

mixed-dimensional 2D/3D HP film to flatten the energy landscape is needed to obviate the detrimental PCB effect.

## Discussion

One would reason that a phase-pure mixed-dimensional film should not suffer from the PCB effect. However, although mixed-dimensional HP films with a narrow $n$ distribution or even a pure phase were previously reported, they were usually determined by bulk characterizations, mainly spectral characterizations. The impurity phase might account for a minor share below the detection limit of these techniques but still effectively hampers the nanoscopic carrier transport[50]. A good illustrative sample is the $CsPbBr_3@Cs_4PbBr_6$ HP system, of which the impurity $CsPbBr_3$ phase is extremely difficult to identify but determines the overall fluorescent property[51]. Besides, high-n 2D HPs with closely similar optical fingerprints to 3D HPs are difficult to identify by spectral characterizations. In this regard, more precise detection means might be required to declare phase-pure 2D HP films.

In summary, this work discloses a PCB mechanism of the promising mixed-dimensional 2D/3D HP system arising from the trap-filling-enhanced built-in potential of the 2D/3D HP interface. It sheds light on the blunt $J_{sc}$ and FF of mixed-dimensional 2D/3D HP solar cells that account for their inferior PCE. The PCB mechanism inspires us to redirect future attention to managing the 2D/3D interfacial traps, and more importantly, reducing the built-in potential of the 2D/3D HP interface instead of focusing on the inherent downsides of 2D HPs.

## Methods
### Chemicals
Phenylethylamine hydroiodide (PEAI, Xi'an Polymer Light Technology Corp.), methyl-ammonium iodide (MAI, 99.5%, Aladdin), $NH_4SCN$ (99.99%, Aladdin), N,N-dimethylformamide (DMF, anhydrous, 99.8%),

dimethyl sulfoxide (DMSO, anhydrous, Aladdin), ethyl alcohol (Standard for GC, >99.8% (GC)), toluene (108-88-3, Sinopharm, controlled agent), diethyl ether (60-29-7, Sinopharm, controlled agent), butyla-mine hydroiodide (BAI, Advanced Election Technology CO., Ltd), octylamine hydroiodide (OAI, Advanced Election Technology CO., Ltd), phenylpropamine hydroiodide (PPAI, Xi'an Polymer Light Technology Corp.). Unless otherwise stated, the reagents and solvents were used directly without any purification.

### Fabrication of the GIHP film
1:1 MAI and $PbI_2$ were dissolved in 5:2 mixed DMF and DMSO to form the precursor with a concentration of 1 M ($Pb^{2+}$). In parallel, the ITO substrates (15 mm × 15 mm) with a channel (with 50 µm channel spacing) were subjected to ultrasonic cleaning in deionized water, acetone, and isopropanol in sequence (10 min each). After that, they were sent to plasma cleaning for 5 min.

The precursor was then filtrated by a filter with an aperture of 0.22 µm. 40 µL of the precursor was deposited on the substrate, and the spin coating was conducted at 5000 rpm for 35 s, the acceleration was 12000 rpm/s. 800 µL of anhydrous toluene as antisolvent was dripped onto the substrate at the 10th second. It should be noted that toluene is the key to obtaining the grain-boundary-slackened film, while the use of diethyl ether would lead to common $MAPbI_3$ films with compact grains. After that, the film was transferred to a hot plate for two-step annealing, first at 60 °C for 1 min and then at 100 °C for 15 min. All the above procedures were conducted in an $N_2$ glovebox.

To prepare the treating solvent, PEAI was dissolved in ethyl alcohol to form a concentration of 2 M. The solution was mixed with a proper amount of toluene until saturation and then filtrated. The $MAPbI_3$ film was immersed in the solution for a proper time. After discontinuing the treatment, the film was quickly washed with toluene to eliminate the residual ethyl alcohol on the surface.

## Fabrication of the phase-graded mixed-dimensional 2D/3D HP films

The following is the fabrication procedure for the $PEA_2MA_4Pb_5I_{16}$ ($n = 5$) film. The glass substrates (15 mm × 15 mm) were subjected to ultrasonic treatment in deionized water, acetone, and isopropanol in sequence for 10 min each. After that, the glass substrates were blown by nitrogen to dry and sent to plasma cleaning for 5 min.

PEAI, MAI, $PbI_2$, and $NH_4SCN$ were dissolved in 1 mL DMF in a molar ratio of 2:4:5:2 to form the precursor with a concentration of 1 M ($Pb^{2+}$). The precursor was then filtrated by a filter with an aperture of 0.22 μm.

The glass substrate was heated on a hot plate at 120 °C for 35 s and rapidly transferred to the spin coater. 35 μL of the precursor was immediately deposited onto the glass substrate followed by spin coating at 5000 rpm for 35 s, and the acceleration was 12000 rpm/s. Subsequently, the film was annealed at 100 °C for 10 min. All the above procedures were conducted in an $N_2$ glovebox.

For the $BA_2MA_4Pb_5I_{16}$ film, BAI, MAI, and $PbI_2$ with a molar ratio of 2:4:5 were dissolved in DMF to form the precursor with a concentration of 1 M ($Pb^{2+}$). The glass substrate was not heated before spin coating; For the $OA_2MA_4Pb_5I_{16}$ film, OAI, MAI, and $PbI_2$ with a molar ratio of 2:4:5 were dissolved in DMF to form the precursor with a concentration of 1 M ($Pb^{2+}$). The glass substrate was heated at 120 °C for 35 s before spin coating; For the $PPA_2MA_4Pb_5I_{16}$ film, PPAI, MAI, $PbI_2$, and $NH_4SCN$ with a molar ratio of 2:4:5:2 were dissolved in 1 mL DMF to form the precursor with a concentration of 1 M ($Pb^{2+}$). The glass substrate was heated at 120 °C for 35 s before spin coating; All the parameters for spin coating and annealing are the same as those of the $PEA_2MA_4Pb_5I_{16}$ film.

## Fabrication of the mixed-dimensional 2D/3D HP films for TA spectrum measurement and time-resolved PL spectrum measurement

The glass substrates (15 mm × 15 mm) were subjected to ultrasonic treatment in deionized water, acetone, and isopropanol in sequence for 10 min each. After that, the glass substrates were blown by nitrogen to dry and sent to plasma cleaning for 5 min. The above four precursors were diluted to 0.1 M with DMF.

The films were fabricated by spin coating, the spin speed was 5000 rpm, the spin time was 35 s, and the acceleration was 12,000 rpm/s. Subsequently, the film was annealed at 100 °C for 10 min. All the above procedures were conducted in an $N_2$ glovebox.

## Fabrication of the ITO/HP/Cu devices for the I-V test

The glass substrates with patterned ITO (12 mm × 12 mm) were subjected to ultrasonic treatment in deionized water, acetone, isopropanol, and deionized water in sequence for 10 min each. After that, the glass substrates were dried in an oven at 60 °C until they were ready. Subsequently, the ITO substrates were sent for ultraviolet ozone cleaning for 10 min.

For the 3D $MAPbI_3$ film, stoichiometric MAI and $PbI_2$ were dissolved in DMF to form a precursor of 1 M, which was filtrated by a filter with an aperture of 0.22 μm. 20 μL of the precursor was deposited onto the ITO substrate for two-step spin coating, first at 1000 rpm for 10 s and then at 5000 rpm for 30 s. The acceleration was 2000 rpm/s. 800 μL of diethyl ether was deposited onto the film as the antisolvent at the 4th second of the second stage of spin coating. After that, the film was annealed at 100 °C for 10 min.

For the $PEA_2MA_{2.67}Pb_{3.67}I_{12}$ film, two precursors were prepared. Precursor-I was fabricated by dissolving stoichiometric reagents ($PbI_2$, MAI, PEAI) to form a stoichiometry of $n = 3$; Precursor-II was fabricated by dissolving stoichiometric reagents to form a stoichiometry of $n = 5$. The concentrations of both precursors were 1 M. They were then mixed with a volume ratio of 2:1 to form Precursor-III, of which the nominal $n$ value is 3.67. The above ITO substrate was

pre-heated at 120 °C for 35 s and rapidly transferred to the spin coater. 20 μL of Precursor-III was deposited onto the ITO substrate, which was followed by one-step spin coating at 5000 rpm for 15 s. The acceleration was 2000 rpm/s. After that, the film was annealed at 120 °C for 10 min.

The $PEA_2(Cs_{1.33}MA_{1.33})Pb_{3.67}I_{12}$ film and the $PEA_2(FA_{1.33}MA_{1.33})Pb_{3.67}I_{12}$ film were fabricated in the same way but only the MAI of Precursor-II was changed to CsI and FAI, respectively. For the $PEA_2(Cs_{0.75}FA_{0.75}MA_{1.5})Pb_4I_{13}$ film, three precursors were prepared: Precursor-I was fabricated by dissolving stoichiometric reagents ($PbI_2$, MAI, PEAI) to form a stoichiometry of $n = 3$; Precursor-II was fabricated by dissolving stoichiometric reagents ($PbI_2$, CsI, PEAI) to form a stoichiometry of $n = 5$; Precursor-III was fabricated by dissolving stoichiometric reagents ($PbI_2$, FAI, PEAI) to form a stoichiometry of $n = 5$; The three precursors were mixed in a volume ratio of 2:1:1 to form Precursor-IV for use. All the other fabrication procedures of the $PEA_2(Cs_{1.33}MA_{1.33})Pb_{3.67}I_{12}$ film, the $PEA_2(FA_{1.33}MA_{1.33})Pb_{3.67}I_{12}$ film, and the $PEA_2(Cs_{0.75}FA_{0.75}MA_{1.5})Pb_4I_{13}$ film were the same as those of the $PEA_2MA_{2.67}Pb_{3.67}I_{12}$ film.

After the spin coating, Cu was deposited onto the films by thermal evaporation. The effective device area is 0.04 $cm^2$. Finally, the devices were sent for the I-V test.

## Characterizations

The TOF-SIMS measurement was conducted by eceshi company (www.eceshi.com) using an IONTOF-TOF SIMS 5 (Germany) system, the detecting area is 500 μm × 500 μm; The depth-profile UPS measurement was carried out by Nanjing Demo Science Co., LTD in a PHI 5000 VersaProbe III system with He I source (21.22 eV), and a negative bias of 9.0 V was applied. The etching was performed by a PHI 5000 VersaProbe III system with an Ar ion sputtering source. The etching area was ~7 $mm^2$, and the sputtering rate was 10 nm/min (calibrated by a standard $SiO_2$ sample); The XRD measurements were carried out with a Bruker D8 Advance XRD system and an Ultima IV (Rigaku) system; The SEM images were obtained by an FEI field emission electron microscope, Quanta 250F; The steady-state PL spectra were measured by a Cary Eclipse Fluorescence Spectrophotometer; The absorption spectra were acquired by a SHIMADZU UV-3600 UV-VIS-NIR spectrophotometer; The transient PL spectra were obtained by a self-built system (ISS PL 1, Champaign, Illinois, USA); The fs-TA spectra and dynamics were recorded using a standard pump-probe configuration at 500 nm, ~100 fs pump pulses at a 1 kHz repetition rate, and a broadband white-light supercontinuum probe (18SI80466 Rev.1, Newport), the spot diameter was 300 μm; The I-V (current-voltage) and I-t (current-time) measurements of the lateral ITO/HP/ITO configuration were obtained by a Keithley 6487 picoammeter. A 532 nm wavelength laser and a 460 nm blue LED were used as the light source; For the I-V measurement of the vertical ITO/HP/Cu configuration, a 425 W collimated Xenon lamp was used as the light source, of which the light intensity was calibrated through a Newport solar simulator to 100 mW $cm^{-2}$ (AM 1.5G). The I-V signals were recorded through a Keithley 2400 source meter.

## Data availability

The source data used in this study are available in the following repository: https://doi.org/10.6084/m9.figshare.21159079. The source data generated in this study are provided in the Source Data file. Source data are provided with this paper.

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

## Acknowledgements

G.X. acknowledges the Science and Technology Development Fund, Macao SAR (File no. FDCT-0044/2020/A1, 0082/2021/A2), UM's research fund (File no. MYRG2020-00151-IAPME), the Natural Science Foundation of China (61935017, 62175268), Guangdong-Hong Kong-Macao Joint Laboratory of Optoelectronic and Magnetic Functional Materials (2019B121205002), and Shenzhen-Hong Kong-Macao Science and Technology Innovation Project (Category C) (SGDX2020110309360100). S. C. acknowledges the financial support from Pengcheng Scholar Program, Shenzhen Science and Technology Program (RCJC20200714114434086, JCYJ20190808142001745, JCYJ20200812160737002, KQTD20160531120 42971), Shenzhen Peacock Plan (No. 20180921273B). D.Y. acknowledges the financial support from China Postdoctoral Science Foundation (No. 2021M702227). We also thank eceshi (www.eceshi.com) for conducting the TOF-SIMS characterization.

## Author contributions

D.Y. conceived the idea, conducted the experiments, and wrote the manuscript; F.C. helped with the experiments and contributed to the manuscript revision with J.L. and B. W.; C.S. and G.X. supervised the project. All authors contributed to scientific discussions.

## Competing interests

The authors declare no competing interests.
