## [Peer Review File · Nature Communications]

Direct Observation of Photoinduced Carrier Blocking in Mixed-Dimensional 2D/3D Perovskites and the OriginReviewers' comments:

Reviewer #1 (Remarks to the Author):

The manuscript discloses a PCB mechanism of the promising mixed dimensional 2D/3D HP system arising from trap filling-enhanced built-in potential of the 2D/3D HP interface. The authors cleverly designed the experimental equipment to prove the working mechanisms. However, for perovskite solar cells, there is a significant difference in carrier transport in the in-plane and out-of-plane directions. Therefore, some conclusions and experiments still need to be further verified and supplemented. Hence, this manuscript is not recommended for publication in this journal, and the detailed comments are as follows:

1. In page 5, the author claimed that “Therefore, the negative impact of the 2D HP fragments, if at all, would be concealed in the ensemble measurement.” However, the failure to explore parallel paths for carrier migration makes the manuscript unconvincing. As reported in a previous article (Nat. commun. 2020, 11, 3308), various types of grain boundaries exist in the perovskite layer, and the effect of charge transport in the in-plane and out-of-plane directions is significantly different. Therefore, the views in the manuscript are hardly convincing.
2. Since the precursor recipe was modified to slacken the grain boundaries, did the authors perform NPC tests for the various precursor recipe to rule out experimental errors?
3. In order to simplify the experimental process, the author selected the MAPbI₃ system as the research object, and it is very important whether the author's views in the manuscript are universal in various systems, such as FAPbI₃. It is worth noting that the change of the system directly affects the carrier diffusion distance. The authors need to rule out whether this factor will have an impact on the views in the manuscript.
4. In page 9, the author claimed that “which suggests a 2D-to-3D charge- or energy transfer.” The direction of the arrows in the schematic diagram can easily lead to misunderstandings by readers, and the author needs to confirm. In addition, the description in the corresponding figure is too complicated, and the author needs to adjust it.
5. In page 17, the author claimed that “The impurity phase might account for a minor share below the detection limit of these techniques but still effectively hampers the nanoscopic carrier transport.” However, recent literature (Adv. Mater. 2021, 34, 2107211) has different views on the high-n phase in 2D perovskites, which identified the high-n phase as an efficient charge transport channel, and the authors need to comment on this.
6. ToC graphic is not strongly relevant to the views in the manuscript, and the authors are advised to make adjustment.

Reviewer #2 (Remarks to the Author):

This work by Yu et al. reported a photoinduced carrier blocking (PCB) mechanism in the 2D/3D hybrid perovskites by applying a combined analysis of physical, optical, and charge transport properties. The experimental techniques are relatively comprehensive. The authors claim that they designed a proof-of-concept prototype where the 2D HPs sufficiently isolate the 3D grains to govern the carrier transport. The results are interesting.

From the schematic diagram of the two-terminal ITO/HP film/ITO device configuration, it is reasonable that a photoinduced carrier blocking (PCB) may exist in the film due to the in-plane charge transport direction in this kind of device structure. However, the transport direction of the charge carrier in solar cells is out-of-the plane direction. Typically, the 2D phases that are parallel to substrate could block the charge transport, whereas the vertical orientated phases will not block the charge transport in perovskite solar cells. The author should make it clear and build connections between their findings and the real solar cells devices before publication. Some detailed comments are given below.

1. How about the I-V characteristics in darkness and under illumination using the device structure of bottom ITO/perovskite/top ITO?
2. Page 4, "They respect a formula of $L2An-1PbnX3n+1$ ". This is a formula of Ruddlesden-Popper (RP) perovskite. There are several 2D perovskite phases, such as Ruddlesden-Popper (RP), Dion-Jacobson (DJ), and alternating cation (ACI) phases.
3. In Figure 3b, the author reported transient PL kinetics of the $n = 2 \sim 4$ 2D phases and the $n = \infty$ 3D phase. The lack of experimental details makes it difficult to understand the discussions in the manuscript. Were the TPL of $n = 2 \sim 4$ 2D phases and the $n = \infty$ 3D phase in Figure 3b measured using the same 2D/3D perovskite film or individual 2D ($n = 2 \sim 4$) and 3D perovskite films? Why the TPL shows an initial PL intensity rise for the 3D HP if the it the pristine 3D perovskite film were measured?
4. Typically, 2D-on-3D bilayer configuration exhibit better optoelectronic performance. The authors claim that the PCB effect is prevented in this kind of perovskite structure. Detailed explanations are suggested to be provided in the manuscript.
5. Page 16, "the PCE of mixed-dimensional 2D/3D HP solar cells could be further improved by increasing the J_{sc} and FF by obviating the PCB effect, as illustrated in Figure 5f". It's not clear that how to obviating the PCB effect in 2D/3D HP solar cells.

Reviewer #3 (Remarks to the Author):

In this manuscript, authors have designed a proof-of-concept prototype where 2D perovskites sufficiently isolate 3D grains to govern the carrier transport, and it was found that 2D perovskites allow free migration of carriers in darkness but block the carrier transport under illumination. Photoinduced carrier blocking (PCB) was proposed and validated through carrier transport and dynamic measurements, and authors have also investigated how does the PCB mechanism reduce the JSC and FF. The results are interesting. But the PCB phenomenon is neither common nor important in most mixed 2D/3D perovskite solar cells with good vertical orientation, unless far excessive 2D perovskites were introduced into 3D grains and with random crystal orientation. Thus, reviewer thinks PCB is not the key limitation for mixed 2D/3D perovskite solar cells, though direct observation of PCB is significant progress. The paper was well organized, and reviewer considers this work could be suitable for publication in Nat Commun after addressing the following issues.

1. Growth mechanism for 3D PVS_K transitioning into 2D PVS_K (Figure 1b) should be experimentally supported and discussed.
2. About the cation exchange-induced growth of 2D perovskites, the time-dependent XRD patterns of GIHP films showed that the intensity of 2D perovskites increases with the decrease of 3D perovskites around 29°, why not appear the signal of < 10°? Any more evidence to support 2D perovskite growth?
3. In Figure S6 (g), the values of energy levels should be marked out for convenient discussion of energy landscape change.
4. In carrier dynamics of the 2D/3D HP interface, the description and discussion on three-stage TA kinetics are not clear. More clear and straightforward results and discussion are needed.

Catalogue

Response to the 1 st Reviewer	2
Response to the 2 nd Reviewer	16
Response to the 3 rd Reviewer.....	25

Reviewer #1 (Remarks to the Author):

The manuscript discloses a PCB mechanism of the promising mixed dimensional 2D/3D HP system arising from trap filling-enhanced built-in potential of the 2D/3D HP interface. The authors cleverly designed the experimental equipment to prove the working mechanisms. However, for perovskite solar cells, there is a significant difference in carrier transport in the in-plane and out-of-plane directions. Therefore, some conclusions and experiments still need to be further verified and supplemented. Hence, this manuscript is not recommended for publication in this journal, and the detailed comments are as follows:

General Response: Dear reviewer, thank you very much for your time and comments. We agree that there is a significant difference between the in-plane and out-of-plane carrier transport in 2D perovskites. In the following (as well as in the revised manuscript), we would like to theoretically and experimentally show you that the photoinduced carrier blocking (PCB) effect also exists in-plane direction. These results would justify the significance of this work for real mixed-dimensional 2D/3D solar cells.

RI_Figure 1. (a) Space diagram of the 2D/3D/2D perovskite configuration in typical mixed-dimensional 2D/3D perovskite films for solar cells. Schematic illustration of the corresponding CBM (b) in darkness and (c) under illumination. (d) Schematic diagram of the ITO/perovskite/Cu configuration for the in-plane direction measurement; Schematic diagram of the corresponding energy landscapes (e) in darkness and (f) under illumination. e represents electrons. ΔV denotes the photoinduced global potential barrier.

The PCB effect stems from the photoinduced energy change at the 2D/3D perovskite interface. From the perspective of energy, as long as the PCB effect is confirmed in the out-of-plane direction, it would naturally occur in the in-plane direction. As schematically illustrated in **RI_Figure 1a-1c**: When the energy band of the 2D perovskite is upshifted relative to the 3D perovskite along the out-of-plane direction, the same band upshift would also occur along the in-plane direction.

Experimentally, a vertical configuration of ITO/perovskite/Cu (RI_Figure 1d) was adopted for measurement. Cu has a close work function to ITO (4.7 eV for Cu and 4.6~4.75 eV for ITO) so that this configuration is almost energetically symmetric the same as the lateral ITO/GIHP film/ITO we presented in the manuscript. It is also an electron-only configuration such that we can exclude interference from bipolar transport. Differently, in this vertical configuration, the in-plane PCB effect could not be confirmed by the NPC phenomenon because the NPC phenomenon is undetectable: On the one hand, the background dark current is so high (~mA level) that the effect of photocurrent is largely masked; On the other hand, the 2D/3D perovskite interface is not likely to be the only carrier path as in the GIHP film, therefore, its NPC effect would be averaged out by other parallel PPC carrier paths.

Alternatively, we could monitor the negative current (I_{neg}) plus positive current (I_{pos}) versus the absolute voltage value (V_{abs}) to gain insights (e.g., I_{neg} at -0.15 eV plus I_{pos} at 0.15 V, the corresponding V_{abs} is 0.15 V). Here, the negative current I_{neg} corresponds to the top-to-bottom electron transport, and the positive current I_{pos} corresponds to the bottom-to-top electron transport. Therefore, the $(I_{\text{neg}}+I_{\text{pos}})$ versus V_{abs} characteristic reflects the symmetry of electron transport. For perfect symmetry, $(I_{\text{neg}}+I_{\text{pos}})$ would be zero for any V_{abs} .

As introduced in the manuscript, the top side of mixed-dimensional 2D/3D perovskite films is rich in the 3D component and the bottom side is rich in the 2D component. In darkness, electrons would freely transport between 2D and 3D perovskites to enable relatively symmetric electron transport (RI_Figure 1e); Under illumination, if the in-plane PCB effect exists, the photo-enhanced built-in potential of the 2D/3D perovskite interface would upshift the energy band of 2D perovskites relative to the 3D perovskite, hence setting a global barrier to the top-to-bottom electron transport (RI_Figure 1f). Therefore, the $(I_{\text{neg}}+I_{\text{pos}})$ versus V_{abs} trends measured in darkness and under illumination should be different.

RI_Figure 2. (a) Two cycles of the I-V measurement of the vertical ITO/PEA₂MA_{2.67}Pb_{3.67}I₁₂/Cu. (a)-(c) are the first-cycle results, (d)-(f) are the second-cycle results. The I-V characteristics measured (a)&(d) in darkness and (b)&(e) under one-sun illumination. (c)&(f) ($I_{neg} + I_{pos}$) versus V_{abs} characteristics in darkness and under illumination.

RI_Figure 3. Two cycles of the I-V measurement of the vertical ITO/MAPbI₃/Cu. (a)-(c) are the first-cycle results, (d)-(f) are the second-cycle results. The I-V characteristics measured (a)&(d) in darkness and (b)&(e) under one-sun illumination. (c)&(f) ($I_{neg} + I_{pos}$) versus V_{abs} characteristics in darkness and under illumination.

RI_Figure 2a-2b reveal a general ohmic contact property the same as the lateral ITO/GIHP film/ITO configuration. As shown in RI_Figure 2c, in darkness, the ($I_{neg} + I_{pos}$) value increases with V_{abs} (more approaching zero). This means the ITO/bottom contact

owns a slightly higher Schottky barrier than the top/Cu contact, therefore, the ITO-to-Cu (bottom-to-top) electron transport is less efficient than the Cu-to-ITO (top-to-bottom) electron transport, but the difference vanishes at high V_{abs} because the Schottky barriers are easy to surmount. Such Schottky barrier disparity agrees with the higher 2D perovskite portion at the bottom of the mixed-dimensional film. In contrast, one-sun illumination engenders a reverse trend of $(I_{neg}+I_{pos})$ versus V_{abs} , *i.e.*, the top-to-bottom electron transport (corresponds to I_{neg}) grows more than the bottom-to-top electron transport (corresponds to I_{pos}) as V_{abs} increases. Indeed, the trend presents a conspicuous diode-like feature that suggests a global photoinduced potential barrier to the top-to-bottom electron transport, which agrees with the effect of the in-plane PCB effect analyzed above. A repeated measurement (RI_Figure 2d-2f) presents the same result, thus excluding experimental errors. However, as a comparison, the vertical ITO/MAPbI₃/Cu configuration is free of such photoinduced change (RI_Figure 3). Its only photoinduced change is the value of $(I_{neg}+I_{pos})$, which is due to photogenerated electrons.

RI_Figure 4. $(I_{neg}+I_{pos})$ versus V_{abs} characteristics of the (a) $PEA_2(Cs_{1.33}MA_{1.33})Pb_{3.67}I_{12}$, (b) $PEA_2(FA_{1.33}MA_{1.33})Pb_{3.67}I_{12}$, and (c) $PEA_2(Cs_{0.67}FA_{0.67}MA_{1.33})Pb_{3.67}I_{12}$ films.

Table RI. Summary of the photoinduced carrier barriers in the various mixed-dimensional 2D/3D perovskite systems ($n= 3.67$)

	A-site cation	MA	$Cs_{0.5}MA_{0.5}$	$FA_{0.5}MA_{0.5}$	$MA_{0.5}Cs_{0.25}&FA_{0.25}$
Photoinduced barrier (meV)	1 st cycle	165.5	204.5	126.8	242.0
	2 nd cycle	91.6	176.8	125.5	224.1

The photoinduced global barrier could further be fitted (by the single exponential function) to be 165.5 meV. Such a high potential barrier far exceeds the thermal perturbation energy (~ 25 meV) such that it would severely hamper electrons and deeply trap holes. Furthermore, the universal presence of the in-plane PCB effect is confirmed in various mixed-dimensional films with different compositions (**RI_Figure 4a-4c**), and the reliability of these results is also supported by repeated measurements. The corresponding photoinduced potential barriers in these films are summarized in **Table RI**. The highest photoinduced barrier reaches even up to 242.0 meV.

In conclusion, the above results confirm the universal presence of the in-plane PCB effect. We believe the PCB effect provides important insights into the compromised J_{sc} and FF of mixed-dimensional 2D/3D perovskite solar cells that jeopardize the PCEs (as analyzed in the manuscript). We do hope these results, alongside the point-by-point responses below, would convince you of the novelty and significance of our work.

I. In page 5, the author claimed that “Therefore, the negative impact of the 2D HP fragments, if at all, would be concealed in the ensemble measurement.” However, the failure to explore parallel paths for carrier migration makes the manuscript unconvincing. As reported in a previous article (Nat. commun. 2020, 11, 3308), various types of grain boundaries exist in the perovskite layer, and the effect of charge transport in the in-plane and out-of-plane directions is significantly different. Therefore, the views in the manuscript are hardly convincing.

Response #1: FIRST, we agree that there could be various carrier paths in common perovskite films. However, to the best of our knowledge, all carrier paths reported show positive photoconductivity (PPC) despite the difference in carrier conductivity. This is also true for the reference recommended (Nat. Commun. 2020, 11, 3308). In this reference, two types of grain boundaries serving as carrier paths are introduced. As clearly stated in the reference, both of them are PPC-type paths. The original text and schematic diagram (**RI_Figure 5**) are presented below:

“...we have developed a unique approach based on tomographic atomic force microscopy, achieving a fully-3D, photogenerated carrier transport map at the nanoscale in hybrid perovskites. *This reveals GBs serving as highly interconnected conducting channels for carrier transport.* We have further discovered the coexistence of *two GB types* in hybrid perovskites, *one exhibiting enhanced carrier mobilities, while the other is insipid...*” (Abstract from Nat. Commun. 2020, 11, 3308)

RI_Figure 5. Proposed models of Type I and Type II GBs. The Y-axis is photocurrent, it is positive for both types of GBs. Adapted from Nat. Commun. 2020, 11, 3308 with notes.

In contrast, our work reports the unique negative photoconductivity (NPC) caused by the energy change of the 2D/3D perovskite interface under illumination. However,

although being one of the prime carrier paths, this NPC-type path is not the only carrier path in common mixed-dimensional 2D/3D perovskite films. Therefore, its negative effect would be averaged out by other parallel PPC-type paths in the ensemble measurement and thus has been unknown to the community. That is why the proof-of-concept GIHP film is designed in this work. In this prototype, the 3D perovskite grains are sufficiently isolated by 2D perovskite flakes at the grain boundaries. As such, the 2D/3D perovskite interface becomes the only carrier path so that its hidden effect would expose.

Therefore, we believe the conclusion of this reference perfectly fits rather than contradicts our discussion. But thanks to your comment, we believe the recommended reference helps strengthen the discussion, and it has been added (alongside another related reference) in the revised manuscript (Page 5):

“...The impact of the 2D HP component on carrier transport is difficult to study in traditional mixed-dimensional 2D/3D HP films because there are complex parallel paths for carrier migration.^{[24]-[25]}...”

(24) Song, J.; Zhou, Y.; Padture, N. P.; Huey, B. D. Anomalous 3D Nanoscale Photoconduction in Hybrid Perovskite Semiconductors Revealed by Tomographic Atomic Force Microscopy. *Nat. Commun.* 2020, **11**, 3308.

(25) Ma, X.; Zhang, F.; Chu, Z.; Hao, J.; Chen, X.; Quan, J.; Huang, Z.; Wang, X.; Li, X.; Yan, Y.; Zhu, K.; Lai, K. Superior Photo-Carrier Diffusion Dynamics in Organic-Inorganic Hybrid Perovskites Revealed by Spatiotemporal Conductivity Imaging. *Nat. Commun.* 2021, **12**, 5009.

SECOND, as for the difference between the in-plane and out-of-plane carrier transport in 2D perovskites, please check the **General Response** above.

2. Since the precursor recipe was modified to slacken the grain boundaries, did the authors perform NPC tests for the various precursor recipe to rule out experimental errors?

Response #2: The key is to use toluene (108-88-3, Sinopharm, controlled agent) as the antisolvent. If it is replaced by diethyl ether (60-29-7, Sinopharm, controlled agent), we would obtain common 3D MAPbI₃ films with compact grains (**RI_Figure 6**, the

right panel is the pristine Figure S1 in the manuscript), and same treatment only results in a disordered film rather than the target GIHP film.

RI_Figure 6. The SEM image of the resultant perovskite film (right panel) by treating a common MAPbI₃ film (left panel, using diethyl ether rather than toluene as the antisolvent) with PEA⁺.

We have emphasized this detail in the experimental section (in the Supporting Information file) as below, and the pristine Figure S1 is changed to RI_Figure 6 above:

“...It should be noted that toluene is the key to obtaining the grain-boundary-slackened film, while the use of diethyl ether would lead to common MAPbI₃ films with compact grains...”

On the other hand, the composition of the proof-of-concept film for the NPC test could not be varied at will, as will be detailed in **Response #3**. But we firmly believe the NPC effect does not result from experimental errors. In fact, we have provided ample experimental evidence, including the I-V measurement (**RI_Figure 7**), the periodic I-t measurement (**RI_Figure 8**), and the excitation intensity-dependent current (**RI_Figure 9**).

Besides, the (in-plane) PCB effect that underlies the NPC phenomenon is confirmed in a series of mixed-dimensional 2D/3D perovskite films with different compositions (see the **General Response**). We believe these measurements confirm the reliability of our work.

RI_Figure 7. (i.e., Figure 2b in the manuscript). I-V characteristics of the GIHP film in darkness and under illumination.

RI_Figure 8 (i.e., Figure S8 in the manuscript). I-t curve of the GIHP film measured under periodic light-on and light-off states. The excitation wavelengths for (a) and (b) are 532 nm and 460 nm, respectively. The applied bias voltage is 0.5 V.

RI_Figure 9 (i.e., Figure S9 in the manuscript). (a) The I-t characteristics of the GIHP film under different excitation intensity levels (@532 nm). The applied bias voltage is 0.5 V. (b) The dependence of photocurrent on the illumination intensity.

3. In order to simplify the experimental process, the author selected the MAPbI₃ system as the research object, and it is very important whether the author's views in the manuscript are universal in various systems, such as FAPbI₃. It is worth noting that the change of the system directly affects the carrier diffusion distance. The authors need to rule out whether this factor will have an impact on the views in the manuscript.

Response #3: With all due respect, we have to make it clear that choosing MAPbI₃ (MA is methylamine) is not for simplification but a must.

To prepare the grain-isolated film, a 3D perovskite film needs to be prepared first, and then it is modified into the GIHP film by a controlled 3D-to-2D phase transition. There are three types of 3D perovskites: CsPbI₃, MAPbI₃, and FAPbI₃ (FA is formamidine). Among them, MAPbI₃ is the only system being thermodynamically stable, while CsPbI₃ and FAPbI₃ are thermodynamically unstable and easily transform into non-perovskite structures.

In contrast, the composition of mixed-dimensional 2D/3D perovskite films for the in-plane PCB effect test could be varied. Because these films are prepared by one-step spin coating, and the 2D structure helps to stabilize the metastable perovskites (*ACS Energy Letters*, 5(6), 1974-1985; *Advanced Materials*, 34(9), 2108556; *Advanced Functional Materials*, 32(15), 2112277). Although FA-pure and Cs-pure films are still challenging to be stabilized, various Cs/FA-involved (50%) films were adopted to confirm the universal presence of the in-plane PCB effect. Please check the **General Response** above.

By the way, we would also like to note that the NPC effect should be independent of the carrier diffusion length. The NPC effect is caused by the photoinduced potential barrier that blocks carriers. It occurs at the 2D/3D perovskite interface, while the carrier diffusion length reflects the carrier transport within the 2D or 3D perovskites.

4. In page 9, the author claimed that “which suggests a 2D-to-3D charge- or energy transfer.” The direction of the arrows in the schematic diagram can easily lead to misunderstandings by readers, and the author needs to confirm. In addition, the description in the corresponding figure is too complicated, and the author needs to adjust it.

Response #4: Thanks for reminding. We have mislabeled the “2D” perovskite and the “3D” perovskite in pristine Figure 3d. It has been corrected as below (**RI_Figure 10**). Besides, we have adjusted the colors to better differentiate the 3D perovskite side (black) and the 2D perovskite side (green).

We have also tried our best to make a more concise version of the discussion as required. The text is too much to present here, please check the revised manuscript (the revisions are highlighted in yellow for easy identification). However, it should also be noted that a comprehensive presentation of the information is necessary. We hope and believe you would understand.

RI_Figure 10. Corrected Figure 3d in the manuscript.

5. In page 17, the author claimed that “The impurity phase might account for a minor share below the detection limit of these techniques but still effectively hampers the nanoscopic carrier transport.” However, recent literature (*Adv. Mater.* 2021, 34, 2107211) has different views on the high- n phase in 2D perovskites, which identified the high- n phase as an efficient charge transport channel, and the authors need to comment on this.

Response #5: With all due respect, we checked the reference you recommended, and the original text reads:

“...**3D-like phases** are more conductive than low-dimensional 2D perovskite phases, due to fewer organic/organic nanoscale interfaces. Nevertheless, abundant **3D-like phases** are intercalated with low-dimensional phases to form an interpenetrating charge transport network throughout the whole film thickness...”

RI_Figure 11. Schematic diagram of the carrier transport in the control sample (left) and the target sample (right) taken from Adv. Mater. 2021, 34, 2107211 with notes.

The authors seemed to mean 3D perovskite rather than high-n phases, which can also be confirmed by the corresponding diagram in the reference (**RI_Figure 11**, marked out by the red dashed circle and the red arrow):

RI_Figure 12. Different comparing systems in our discussion and the reference recommended.

On the other hand, as presented in **RI_Figure 12**, our discussion and the reference refer to different comparing systems. The reference compares, in the very same mixed-

dimensional 2D/3D perovskite film, the low-n phase channel (the carrier channel passing through low-n phases) and the 3D-like phase channel, while our discussion globally compares the 3D perovskite films with and without the 2D phases (no matter high-n ones or low-n ones).

Therefore, we do not think the reference recommended contradicts our discussion. On the contrary, we believe the reference agrees with our opinion. It also exactly shows the significance of our work: While we all know that the 2D perovskite components hamper the carrier transport, the reason is still unclear. The PCB mechanism we unveil is thus of fundamental importance.

6. ToC graphic is not strongly relevant to the views in the manuscript, and the authors are advised to make adjustment.

Response #6: With all due respect, while the ToC graphic is presented with some art style, it exactly presents the idea of our work. The details are presented in **RI_Figure I3:**

RI_Figure I3. Details of the ToC graphic and how it is related to the theme of our work.

Our work introduces the photoinduced barrier at the 2D/3D perovskite interface to block the carriers. The basic components in the ToC graphic are 2D perovskite, 3D perovskite, light, the photoinduced barrier of the 2D/3D perovskite interface, a freely moving carrier, a blocked carrier, and the device structure of typical 2D/3D perovskite solar cells. The assembly of them well reflects our idea. We do hope you agree, and we sincerely welcome your further suggestions on the ToC graphic after reviewing the

above illustration.

In the end, we would like to thank you again for your valuable time and comments. We welcome different views for improving this work in every measure. We sincerely hope the above response addresses your concerns and justifies the publication of this manuscript in *Nature Communications*. Any further comments are also highly welcomed.

Reviewer #2 (Remarks to the Author):

This work by Yu et al. reported a photoinduced carrier blocking (PCB) mechanism in the 2D/3D hybrid perovskites by applying a combined analysis of physical, optical, and charge transport properties. The experimental techniques are relatively comprehensive. The authors claim that they designed a proof-of-concept prototype where the 2D HPs sufficiently isolate the 3D grains to govern the carrier transport. The results are interesting.

From the schematic diagram of the two-terminal ITO/HP film/ITO device configuration, it is reasonable that a photoinduced carrier blocking (PCB) may exist in the film due to the in-plane charge transport direction in this kind of device structure. However, the transport direction of the charge carrier in solar cells is out-of-the plane direction. Typically, the 2D phases that are parallel to substrate could block the charge transport, whereas the vertical orientated phases will not block the charge transport in perovskite solar cells. The author should make it clear and build connections between their findings and the real solar cells devices before publication. Some detailed comments are given below.

General Response: Dear reviewer, thank you very much for your time and comments. We believe you mean that we presented the photoinduced carrier blocking (PCB) in the “out-of-plane direction” of 2D perovskites, while real mixed-dimensional 2D/3D perovskite solar cells transport carriers along the “in-plane” direction.

First, we would like to theoretically and experimentally show you that the photoinduced carrier blocking (PCB) effect also exists in-plane direction. These results would justify the significance of this work for real mixed-dimensional 2D/3D solar cells.

R2_Figure 1. (a) Space diagram of the 2D/3D/2D perovskite configuration in typical mixed-dimensional 2D/3D perovskite films for solar cells. Schematic illustration of the corresponding CBM (b) in darkness and (c) under illumination. (d) Schematic diagram of the ITO/perovskite/Cu configuration for the in-plane direction measurement; Schematic diagram of the corresponding energy landscapes (e) in darkness and (f) under illumination. e represents electrons. ΔV denotes the photoinduced global potential barrier.

The PCB effect stems from the photoinduced energy change at the 2D/3D perovskite interface. From the perspective of energy, as long as the PCB effect is confirmed in the out-of-plane direction, it would naturally occur in the in-plane direction. As schematically illustrated in **R2_Figure 1a-1c**: When the energy band of the 2D perovskite is upshifted relative to the 3D perovskite along the out-of-plane direction, the same band upshift would also occur along the in-plane direction.

Experimentally, a vertical configuration of ITO/perovskite/Cu (**R2_Figure 1d**) was adopted for measurement. Cu has a close work function to ITO (4.7 eV for Cu and 4.6~4.75 eV for ITO) so that this configuration is almost energetically symmetric the same as the lateral ITO/GIHP film/ITO we presented in the manuscript. It is also an electron-only configuration such that we can exclude interference from bipolar transport. Differently, in this vertical configuration, the in-plane PCB effect could not be confirmed by the NPC phenomenon because the NPC phenomenon is undetectable: On the one hand, the background dark current is so high (~mA level) that the effect of photocurrent is largely masked; On the other hand, the 2D/3D perovskite interface is not likely to be the only carrier path as in the GIHP film, therefore, its NPC effect would be averaged out by other parallel PPC carrier paths.

Alternatively, we could monitor the negative current (I_{neg}) plus positive current (I_{pos}) versus the absolute voltage value (V_{abs}) to gain insights (e.g., I_{neg} at -0.15 eV plus I_{pos} at 0.15 V, the corresponding V_{abs} is 0.15 V). Here, the negative current I_{neg} corresponds to the top-to-bottom electron transport, and the positive current I_{pos} corresponds to the bottom-to-top electron transport. Therefore, the $(I_{\text{neg}}+I_{\text{pos}})$ versus V_{abs} characteristic reflects the symmetry of electron transport. For perfect symmetry, $(I_{\text{neg}}+I_{\text{pos}})$ would be zero for any V_{abs} .

As introduced in the manuscript, the top side of mixed-dimensional 2D/3D perovskite films is rich in the 3D component and the bottom side is rich in the 2D component. In darkness, electrons would freely transport between 2D and 3D perovskites to enable relatively symmetric electron transport (R2_Figure 1e); Under illumination, if the in-plane PCB effect exists, the photo-enhanced built-in potential of the 2D/3D perovskite interface would upshift the energy band of 2D perovskites relative to the 3D perovskite, hence setting a global barrier to the top-to-bottom electron transport (R2_Figure 1f). Therefore, the $(I_{\text{neg}}+I_{\text{pos}})$ versus V_{abs} trends measured in darkness and under illumination should be different.

R2_Figure 2. (a) Two cycles of the I-V measurement of the vertical ITO/PEA₂MA_{2.67}Pb_{3.67}I₁₂/Cu. (a)-(c) are the first-cycle results, (d)-(f) are the second-cycle results. The I-V characteristics measured (a)&(d) in darkness and (b)&(e) under one-sun illumination. (c)&(f) $(I_{\text{neg}} + I_{\text{pos}})$ versus V_{abs} characteristics in darkness and under illumination.

R2_Figure 3. Two cycles of the I-V measurement of the vertical ITO/MAPbI₃/Cu. (a)-(c) are the first-cycle results, (d)-(f) are the second-cycle results. The I-V characteristics measured (a)&(d) in darkness and (b)&(e) under one-sun illumination. (c)&(f) ($I_{neg} + I_{pos}$) versus V_{abs} characteristics in darkness and under illumination.

R2_Figure 2a-2b reveal a general ohmic contact property the same as the lateral ITO/GIHP film/ITO configuration. As shown in R2_Figure 2c, in darkness, the ($I_{neg} + I_{pos}$) value increases with V_{abs} (more approaching zero). This means the ITO/bottom contact owns a slightly higher Schottky barrier than the top/Cu contact, therefore, the ITO-to-Cu (bottom-to-top) electron transport is less efficient than the Cu-to-ITO (top-to-bottom) electron transport, but the difference vanishes at high V_{abs} because the Schottky barriers are easy to surmount. Such Schottky barrier disparity agrees with the higher 2D perovskite portion at the bottom of the mixed-dimensional film. In contrast, one-sun illumination engenders a reverse trend of ($I_{neg} + I_{pos}$) versus V_{abs} , i.e., the top-to-bottom electron transport (corresponds to I_{neg}) grows with V_{abs} more than the bottom-to-top electron transport (corresponds to I_{pos}). Indeed, the trend presents a conspicuous diode-like feature that suggests a global photoinduced potential barrier to the top-to-bottom electron transport, which agrees with the effect of the in-plane PCB effect analyzed above. A repeated measurement (R2_Figure 2d-2f) presents the same result, thus excluding experimental errors. However, as a comparison, the vertical ITO/MAPbI₃/Cu configuration is free of such photoinduced change (**R2_Figure 3**). Its only photoinduced change is the value of ($I_{neg} + I_{pos}$), which is due to photogenerated electrons.

R2_Figure 4. ($I_{\text{neg}} + I_{\text{pos}}$) versus V_{abs} characteristics of the (a) $\text{PEA}_2(\text{Cs}_{1.33}\text{MA}_{1.33})\text{Pb}_{3.67}\text{I}_{12}$, (b) $\text{PEA}_2(\text{FA}_{1.33}\text{MA}_{1.33})\text{Pb}_{3.67}\text{I}_{12}$, and (c) $\text{PEA}_2(\text{Cs}_{0.67}\text{FA}_{0.67}\text{MA}_{1.33})\text{Pb}_{3.67}\text{I}_{12}$ films.

Table R2. Summary of the photoinduced carrier barriers in the various mixed-dimensional 2D/3D perovskite systems ($n = 3.67$)

		A-site cation	MA	$\text{Cs}_{0.5}\text{MA}_{0.5}$	$\text{FA}_{0.5}\text{MA}_{0.5}$	$\text{MA}_{0.5}\text{Cs}_{0.25}\&\text{FA}_{0.25}$
Photoinduced barrier (meV)	1 st cycle		165.5	204.5	126.8	242.0
	2 nd cycle		91.6	176.8	125.5	224.1

The photoinduced global barrier could further be fitted (by the single exponential function) to be 165.5 meV. Such a high potential barrier far exceeds the thermal perturbation energy (~25 meV) such that it would severely hamper electrons and deeply trap holes. Furthermore, the universal presence of the in-plane PCB effect is confirmed in various mixed-dimensional films with different compositions (**R2_Figure 4a-4c**), and the reliability of these results is also supported by repeated measurements. The corresponding photoinduced potential barriers in these films are summarized in **Table R2**. The highest photoinduced barrier reaches even up to 242.0 meV.

In conclusion, the above results confirm the universal presence of the in-plane PCB effect. We believe the PCB effect provides important insights into the compromised J_{sc} and FF of mixed-dimensional 2D/3D perovskite solar cells that jeopardize the PCEs

(as analyzed in the manuscript). We do hope these results, alongside the point-by-point responses below, would convince you of the novelty and significance of our work.

1. How about the I-V characteristics in darkness and under illumination using the device structure of bottom ITO/perovskite/top ITO?

Response #1: We have measured the I-V characteristics of a similar ITO/perovskite/Cu configuration. Cu has a closely similar work function with ITO so that ITO/perovskite/Cu works similarly to the ITO/perovskite/top ITO configuration you required. The details can be found in the **General Response** above. Please check.

2. Page 4, "They respect a formula of $L_2A_{n-1}Pb_nX_{3n+1}$ ". This is a formula of Ruddlesden-Popper (RP) perovskite. There are several 2D perovskite phase, such as Ruddlesden-Popper (RP), Dion-Jacobson (DJ), and alternating cation (ACI) phases.

Response #2: Thanks for reminding. Indeed, 2D perovskites could be divided into $\langle 100 \rangle$, $\langle 110 \rangle$, and $\langle 111 \rangle$ types, and the $\langle 100 \rangle$ type could be further divided into the Ruddlesden-Popper type, the Dion-Jacobson type, the Aurivillius type, and the alternating cations in the interlayer (ACI) type. Therefore, we have made proper revisions as below (Page 4 in the manuscript, highlighted in yellow):

"...2D HPs can be viewed as partitioned 3D HPs by bulky ligands, and they could be subcategorized as $\langle 100 \rangle$, $\langle 110 \rangle$, and $\langle 111 \rangle$ -oriented types depending on the partitioning manner.^[6] The most widely studied Ruddlesden-Popper type (belonging to the $\langle 100 \rangle$ type) respects a formula of $L_2A_{n-1}Pb_nX_{3n+1}$, where L is an organic ligand; A is methylammonium (MA), formamidinium (FA), or Cs; B is Pb or Sn; X is a halogen Cl, Br, or I; n denotes the HP layer number between two ligand layers..."

(6) Mitzi, D. B. Templating and Structural Engineering in Organic-Inorganic Perovskites. *J. Chem. Soc., Dalton Trans.* 2001, **null**, 1-12.

3. In Figure 3b, the author reported transient PL kinetics of the $n=2\sim 4$ 2D phases and the $n=\infty$ 3D phase. The lack of experimental details makes it difficult to understand the discussions in the manuscript. Were the TPL of $n=2\sim 4$ 2D phases and the $n=\infty$ 3D phase in Figure 3b measured using the same 2D/3D perovskite film or individual 2D ($n=2\sim 4$) and 3D perovskite films? Why the TPL shows an initial PL intensity rise for the 3D HP if the it the pristine 3D perovskite film were measured?

Response #3: Yes, the TRPL spectra of the $n=2\sim 4$ 2D phases and the $n=\infty$ 3D phase in Figure 3b were measured in the same mixed-dimensional 2D/3D perovskite film, and it's the same film for the TA measurement.

The TRPL spectrum measurement was carried out to understand the stage ① in the TA spectra, where the PB decay of the 2D perovskite phases is accompanied by the PB rise of the 3D perovskite phase. This phenomenon can be caused by either carrier transfer or energy transfer (i.e., electron-hole pair transfer). The TRPL spectra reveal an initial signal rise of the 3D perovskite phase, thus confirming that stage ① is an energy transfer process rather than a carrier transfer process.

We have made it clear in the revised manuscript as below (Page 10 in the revised manuscript):

“...The first stage (labeled as ①) reveals rapid photobleaching (PB) decay for the 2D HPs but a PB rise for the 3D HP, which suggests a 2D-to-3D charge- or energy transfer. The corresponding time-resolved PL spectrum (Figure 3b) shows an initial PL intensity rise for the 3D HP but a decay for the 2D ones, and the feature time closely coincides with that of ①. Therefore, ① should be (mainly) ascribed to the well-known Förster energy transfer from wide-bandgap 2D phases to the narrow-bandgap 3D phase...”

4. Typically, 2D-on-3D bilayer configuration exhibit better optoelectronic performance. The authors claim that the PCB effect is prevented in this kind of perovskite structure. Detailed explanations are suggested to be provided in the manuscript.

Response #4: In a mixed-dimensional 2D/3D perovskite solar cell, the 2D perovskite components intersperse in the 3D perovskite matrix so that the resultant 3D/2D/2D double heterojunction becomes one of the primes carrier paths. The PCB effect makes such double heterojunctions an energy barrier to hamper the carriers (electrons being blocked and holes being trapped), thus decreasing the J_{sc} and FF of the solar cell; The 2D-on-3D bilayer configuration is essentially a unidirectional 2D/3D perovskite heterojunction, and the built-in potential enhancement of the heterojunction upon illumination only helps to separate the electrons and holes, which increases the charge collection efficiency of solar cells.

We have intensified the discussion in the revised manuscript as below (Page 20 in

the revised manuscript):

“...Noteworthy, the PCB mechanism also justifies the usually better optoelectronic performance of the 2D-on-3D bilayer configuration.^{[56]-[59]} Because the 2D-on-3D bilayer configuration is essentially a unidirectional 2D/3D HP heterojunction, the photo-enhanced built-in potential of the heterojunction only helps to separate the electrons and holes, thus improving the charge collection efficiency of the solar cell rather than causing a detrimental PCB effect as in the mixed-dimensional 2D/3D HP counterpart...”

5. Page 16, “the PCE of mixed-dimensional 2D/3D HP solar cells could be further improved by increasing the J_{sc} and FF by obviating the PCB effect, as illustrated in Figure 5f”. It’s not clear that how to obviating the PCB effect in 2D/3D HP solar cells.

Response #5: The PCB effect originates from the built-in potential of the 2D/3D perovskite interface, and the built-in potential results from the inconsistent Fermi levels of the 2D perovskite phase and the 3D perovskite phase. Therefore, manipulation of the (self-) doping across the mixed-dimensional film is required to flatten the energy landscape in order to obviate the PCB effect. In the pristine manuscript, this part of discussion is presented at the end of the conclusion section, now it is moved to the part you mentioned as below (Page 20 in the revised manuscript):

“...Accordingly, we reason that the PCE of mixed-dimensional 2D/3D HP solar cells could be further improved by increasing the J_{sc} and FF by obviating the PCB effect, as illustrated in Figure 7e. The built-in potential stems from the inconsistent (self-) doping levels of the 2D and 3D HP components. Therefore, delicate doping across the mixed-dimensional 2D/3D HP film to flatten the energy landscape is needed to obviate the detrimental PCB effect...”

In the end, we would like to thank you again for your valuable time and helpful professional comments. We sincerely hope the above response addresses your concerns and justifies the publication of this manuscript in *Nature Communications*. Any further comments are also highly welcomed.

Reviewer #3 (Remarks to the Author):

In this manuscript, authors have designed a proof-of-concept prototype where 2D perovskites sufficiently isolate 3D grains to govern the carrier transport, and it was found that 2D perovskites allow free migration of carriers in darkness but block the carrier transport under illumination. Photoinduced carrier blocking (PCB) was proposed and validated through carrier transport and dynamic measurements, and authors have also investigated how does the PCB mechanism reduce the J_{SC} and FF. The results are interesting. But the PCB phenomenon is neither common nor important in most mixed 2D/3D perovskite solar cells with good vertical orientation, unless far excessive 2D perovskites were introduced into 3D grains and with random crystal orientation. Thus, reviewer thinks PCB is not the key limitation for mixed 2D/3D perovskite solar cells, though direct observation of PCB is significant progress. The paper was well organized, and reviewer considers this work could be suitable for publication in Nat Commun after addressing the following issues.

General Response: Dear reviewer, thank you very much for your time and comments. First, we would like to theoretically and experimentally show you that the photoinduced carrier blocking (PCB) effect also exists in-plane direction. Based on these results, we firmly believe the PCB effect is a vital factor that limits the performance of mixed-dimensional 2D/3D perovskite solar cells.

R3_Figure 1. (a) Space diagram of the 2D/3D/2D perovskite configuration in typical

mixed-dimensional 2D/3D perovskite films for solar cells. Schematic illustration of the corresponding CBM (b) in darkness and (c) under illumination. (d) Schematic diagram of the ITO/perovskite/Cu configuration for the in-plane direction measurement; Schematic diagram of the corresponding energy landscapes (e) in darkness and (f) under illumination. e represents electrons. ΔV denotes the photoinduced global potential barrier.

The PCB effect stems from the photoinduced energy change at the 2D/3D perovskite interface. From the perspective of energy, as long as the PCB effect is confirmed in the out-of-plane direction, it would naturally occur in the in-plane direction. As schematically illustrated in **R3_Figure 1a-1c**: When the energy band of the 2D perovskite is upshifted relative to the 3D perovskite along the out-of-plane direction, the same band upshift would also occur along the in-plane direction.

Experimentally, a vertical configuration of ITO/perovskite/Cu (R3_Figure 1d) was adopted for measurement. Cu has a close work function to ITO (4.7 eV for Cu and 4.6~4.75 eV for ITO) so that this configuration is almost energetically symmetric the same as the lateral ITO/GIHP film/ITO we presented in the manuscript. It is also an electron-only configuration such that we can exclude interference from bipolar transport. Differently, in this vertical configuration, the in-plane PCB effect could not be confirmed by the NPC phenomenon because the NPC phenomenon is undetectable: On the one hand, the background dark current is so high (~mA level) that the effect of photocurrent is largely masked; On the other hand, the 2D/3D perovskite interface is not likely to be the only carrier path as in the GIHP film, therefore, its NPC effect would be averaged out by other parallel PPC carrier paths.

Alternatively, we could monitor the negative current (I_{neg}) plus positive current (I_{pos}) versus the absolute voltage value (V_{abs}) to gain insights (e.g., I_{neg} at -0.15 eV plus I_{pos} at 0.15 V, the corresponding V_{abs} is 0.15 V). Here, the negative current I_{neg} corresponds to the top-to-bottom electron transport, and the positive current I_{pos} corresponds to the bottom-to-top electron transport. Therefore, the $(I_{neg}+I_{pos})$ versus V_{abs} characteristic reflects the symmetry of electron transport. For perfect symmetry, $(I_{neg}+I_{pos})$ would be zero for any V_{abs} .

As introduced in the manuscript, the top side of mixed-dimensional 2D/3D perovskite films is rich in the 3D component and the bottom side is rich in the 2D component. In darkness, electrons would freely transport between 2D and 3D perovskites to enable relatively symmetric electron transport (R3_Figure 1e); Under illumination, if the in-plane PCB effect exists, the photo-enhanced built-in potential of

the 2D/3D perovskite interface would upshift the energy band of 2D perovskites relative to the 3D perovskite, hence setting a global barrier to the top-to-bottom electron transport (R3_Figure 1f). Therefore, the ($I_{\text{neg}}+I_{\text{pos}}$) versus V_{abs} trends measured in darkness and under illumination should be different.

R3_Figure 2. (a) Two cycles of the I-V measurement of the vertical ITO/PEA₂MA_{2.67}Pb_{3.67}I₁₂/Cu. (a)-(c) are the first-cycle results, (d)-(f) are the second-cycle results. The I-V characteristics measured (a)&(d) in darkness and (b)&(e) under one-sun illumination. (c)&(f) ($I_{\text{neg}} + I_{\text{pos}}$) versus V_{abs} characteristics in darkness and under illumination.

R3_Figure 3. Two cycles of the I-V measurement of the vertical ITO/MAPbI₃/Cu. (a)-(c) are the first-cycle results, (d)-(f) are the second-cycle results. The I-V characteristics measured (a)&(d) in darkness and (b)&(e) under one-sun illumination. (c)&(f) ($I_{\text{neg}} + I_{\text{pos}}$) versus V_{abs} characteristics in darkness and under illumination.

R3_Figure 2a-2b reveal a general ohmic contact property the same as the lateral ITO/GIHP film/ITO configuration. As shown in **R3_Figure 2c**, in darkness, the $(I_{\text{neg}}+I_{\text{pos}})$ value increases with V_{abs} (more approaching zero). This means the ITO/bottom contact owns a slightly higher Schottky barrier than the top/Cu contact, therefore, the ITO-to-Cu (bottom-to-top) electron transport is less efficient than the Cu-to-ITO (top-to-bottom) electron transport, but the difference vanishes at high V_{abs} because the Schottky barriers are easy to surmount. Such Schottky barrier disparity agrees with the higher 2D perovskite portion at the bottom of the mixed-dimensional film. In contrast, one-sun illumination engenders a reverse trend of $(I_{\text{neg}}+I_{\text{pos}})$ versus V_{abs} , i.e., the top-to-bottom electron transport (corresponds to I_{neg}) grows with V_{abs} more than the bottom-to-top electron transport (corresponds to I_{pos}). Indeed, the trend presents a conspicuous diode-like feature that suggests a global photoinduced potential barrier to the top-to-bottom electron transport, which agrees with the effect of the in-plane PCB effect analyzed above. A repeated measurement (**R3_Figure 2d-2f**) presents the same result, thus excluding experimental errors. However, as a comparison, the vertical ITO/MAPbI₃/Cu configuration is free of such photoinduced change (**R3_Figure 3**). Its only photoinduced change is the value of $(I_{\text{neg}}+I_{\text{pos}})$, which is due to photogenerated electrons.

R3_Figure 4. ($J_{\text{neg}} + J_{\text{pos}}$) versus V_{abs} characteristics of the (a) $\text{PEA}_2(\text{Cs}_{1.33}\text{MA}_{1.33})\text{Pb}_{3.67}\text{I}_{12}$, (b) $\text{PEA}_2(\text{FA}_{1.33}\text{MA}_{1.33})\text{Pb}_{3.67}\text{I}_{12}$, and (c) $\text{PEA}_2(\text{Cs}_{0.67}\text{FA}_{0.67}\text{MA}_{1.33})\text{Pb}_{3.67}\text{I}_{12}$ films.

Table R3. Summary of the photoinduced carrier barriers in the various mixed-dimensional 2D/3D perovskite systems ($n = 3.67$)

		A-site cation	MA	$\text{Cs}_{0.5}\text{MA}_{0.5}$	$\text{FA}_{0.5}\text{MA}_{0.5}$	$\text{MA}_{0.5}\text{Cs}_{0.25}\&\text{FA}_{0.25}$
Photoinduced barrier (meV)	1 st cycle		165.5	204.5	126.8	242.0
	2 nd cycle		91.6	176.8	125.5	224.1

The photoinduced global barrier could further be fitted (by the single exponential function) to be 165.5 meV. Such a high potential barrier far exceeds the thermal perturbation energy (~25 meV) such that it would severely hamper electrons and deeply trap holes. Furthermore, the universal presence of the in-plane PCB effect is confirmed in various mixed-dimensional films with different compositions (**R3_Figure 4a-4c**), and the reliability of these results is also supported by repeated measurements. The corresponding photoinduced potential barriers in these films are summarized in **Table R3**. The highest photoinduced barrier reaches even up to 242.0 meV.

In conclusion, the above results confirm the universal presence of the in-plane PCB effect. We believe the PCB effect provides important insights into the compromised J_{sc} and FF of mixed-dimensional 2D/3D perovskite solar cells that jeopardize the PCEs

(as analyzed in the manuscript). We do hope these results, alongside the point-by-point responses below, would convince you of the novelty and significance of our work.

I. Growth mechanism for 3D PVSK transitioning into 2D PVSK (Figure 1b) should be experimentally supported and discussed.

Response #1: With all due respect, we have discussed the 3D-to-2D phase transition mechanism in detail (the second paragraph on Page 6, also presented below) with ample experimental support (the PL spectrum characterization, SEM characterization, and XRD characterization). 2D perovskites are essentially partitioned 3D perovskites by bulky organic spacers. Therefore, the treatment with a solution containing organic spacers would drive the 3D-to-2D phase transition. In literature, it usually takes place on the surface of the 3D perovskite film, but we controlled the transitioning sites to be the grain boundaries:

“...The transition course was recorded by the treating time-dependent scanning electron microscope (SEM) characterization (Figure S2): In the grain boundaries, platelet-shaped crystals that accord with the structure of 2D HPs emerge and gradually grow over time. The one-dimensional X-ray diffraction (XRD) characterization shows the nucleation and progressive strengthening of the 2D HP peaks (Figure S3) in line with the growth of the platelet crystals. Consistently, the photoluminescence (PL) spectra (Figure 1c) reveal the appearance of the characteristic peaks of 2D HPs and, concurrently, a waning relative peak intensity of the 3D HP, which confirms the 3D-to-2D HP transition. A slight blueshift of the 3D HP PL peak can be observed after the treatment, agreeing with the decrease in grain size caused by the edge consumption. Therefore, we can obtain an as-designed HP film with the connection between 3D HP grains sufficiently, if not entirely, cut by 2D HP platelets in the grain boundaries...”

R3_Figure 5 (i.e., Figure S2 in the manuscript). (a)-(d) SEM micrographs of the GIHP film after different treating times. The scale bars in a-d are 5 μm .

R3_Figure 6 (i.e., Figure S3 in the manuscript). (a) The treating time-dependent XRD pattern of the GIHP film. # denotes the diffraction peak of 2D perovskites, while * denotes the diffraction peaks of the parent 3D MAPbI₃. (b) A zoomed view that shows the growing diffraction intensity of 2D perovskites and the concurrent waning diffraction intensity of the 3D perovskites upon increasing treating time. Besides, the 2D perovskite diffraction peak shifts towards the smaller angle region, indicating the decrease of the average n value (quantum well width) that reduces the interlayer distance.^[2]

R3_Figure 7 (i.e., Figure 1c in the manuscript). Treating time-dependent PL spectra of the film.

2. About the cation exchange-induced growth of 2D perovskites, the time-dependent XRD patterns of GIHP films showed that the intensity of 2D perovskites increases with the decrease of 3D perovskites around 29°, why not appear the signal of < 10°? Any more evidence to support 2D perovskite growth?

Response #2: According to Bragg's law, the <10° diffraction could be caused by facets with a large interlayer spacing, namely the (00l) facets of 2D perovskites (the nominal

of the facets are parallel to the out-of-plane direction). Therefore, only when the (00 l) facets of 2D perovskites are parallel to the substrate, the $<10^\circ$ peaks would emerge. However, in this work, the 2D perovskites are vertical to the substrate, and only (100) and (010) planes could be identified. These facets have similar interplanar spacing to that of the (00 l) facets of the 3D perovskite phase (**R3_Figure 8**). The diffraction peak marked with * (see R3_Figure 6) is the (002) peak of 3D MAPbI₃, and the diffraction peak marked with # belongs to the (100) or (010) facets of the 2D perovskite phases. The slight difference could be ascribed to the tilting of the [PbI₆] octahedra. That's why the marked two peaks are both around 29°.

R3_Figure 8. Schematic diagram of (a) the (00 l) / (010) facets of 2D perovskites and (b) the (00 l) facets of 3D perovskites.

We are afraid the current characterizations are the best we can present, but we believe the PL, SEM, and XRD characterizations powerfully confirm the 2D-to-3D phase transition. They visualize the transition in three different aspects: optical properties, spatial location, and lattice structure. The combination of them strongly supports the growth of 2D perovskites at the boundaries of 3D perovskite grains.

3. In Figure S6 (g), the values of energy levels should be marked out for convenient discussion of energy landscape change.

Response #3: The values are not marked out for the following reason: The band energy values are determined by depth-profile UPS in phase-graded mixed dimensional 2D/3D perovskite films. As illustrated in the manuscript, despite the global dimensionality reduction with depth, each cross section comprises mixed phases with different n values. Therefore, the UPS results could correctly reflect the energy band variation with the n value (so that we can picture the band bending of the 2D/3D perovskite interface for discussion) but inevitably deviate from the practical values of phase-pure 2D perovskites with different n values. Although not precise enough, the results are correct enough for the discussion.

Noteworthy, some previous reports may measure the band energy independently on phase-pure 2D and 3D perovskites, and then combine the results to mimic the band alignment of the 2D/3D perovskite heterojunction. This method might cause significant deviations, because the interface properties that could significantly influence the Fermi levels are greatly different in the isolated state and the contact state. That's why the in-situ depth-profile UPS measurement is adopted in this work.

4. In carrier dynamics of the 2D/3D HP interface, the description and discussion on three-stage TA kinetics are not clear. More clear and straightforward results and discussion are needed.

Response #4: We have also tried our best to make a clearer and more concise version of the discussion as required. The text is too much to present here, please check the revised manuscript (the revisions are highlighted in yellow for easy identification). However, it should also be noted that a comprehensive presentation of the information is necessary. We hope and believe you would understand.

In the end, we would like to thank you again for your valuable time and helpful professional comments. We sincerely hope the above response addresses your concerns and justifies the publication of this manuscript in *Nature Communications*. Any further comments are also highly welcomed.

REVIEWER COMMENTS

Reviewer #1 (Remarks to the Author):

In Figure 6 and the associated description, the authors need to reconfirm the accuracy of the arrows and description in in-plane and out-of-plane directions. Overall, the authors provide detailed experimental supplements and discussions of previous issues that address concerns about the previous manuscript, therefore I recommend the revised manuscript for publication.

Reviewer #2 (Remarks to the Author):

The authors have tried to address the concerns of the reviewer. However, some concerns have not been fully addressed. These comments need to be addressed before publication. For example,

1. The in plane (organic layer parallel to the substrate) and out of plane (organic layer vertical to the substrate) directions of 2D perovskites are not fully understood.
2. The authors claim that the photoinduced carrier blocking (PCB) effect exists both in-plane and out plane direction in 2D perovskites. For mixed dimensional 2D/3D HP solar cells, the authors discussed in the manuscript as follows: "Although freely moving in darkness, electrons would be greatly hampered under illumination, and holes would be also deeply trapped." While for the 2D-on-3D bilayer configuration, the authors don't think the PCB effect play the roles.
3. The authors claim that ITO/HP/Cu is also an electron-only configuration, which excludes the interference from bipolar transport. Why? Typically, electron transport layer (ETL) should be included in configuration, for example, ITO/ETL/HP/ETL/Cu.

Reviewer #3 (Remarks to the Author):

The authors have already addressed most of the issues from the reviewers, and the paper could be accepted with this version.

Catalogue

Response to the 1 st Reviewer	2
Response to the 2 nd Reviewer	7
Response to the 3 rd Reviewer.....	16

Reviewer #1 (Remarks to the Author):

In Figure 6 and the associated description, the authors need to reconfirm the accuracy of the arrows and description in in-plane and out-of-plane directions. Overall, the authors provide detailed experimental supplements and discussions of previous issues that address concerns about the previous manuscript, therefore I recommend the revised manuscript for publication.

Response: Dear reviewer, thank you very much for your time and comments. We recognize that sometimes the so-called “plane” of the “in-plane direction” and the “out-of-plane direction” refers to the substrate (such that the “in-plane direction” denotes a direction parallel to the substrate and the “out-of-plane direction” denotes a direction normal to the substrate). However, as far as we know, this definition is more often adopted for 3D HP-based vertical devices (as shown in RI_Figure 1),^[1-2] because 3D HPs have a quasi-symmetric structure, and a direction could only be indicated by referring to the substrate.

RI_Figure 1. The definitions of “in-plane” and “out-of-plane” directions in 3D HP films. X-ray diffraction in the (a) “out-of-plane” direction and (b) the “in-plane” direction. This figure is taken from the reference *Sci. Adv.* **3**, eaao5616 (2017) (Figure 1c and 1d).^[1]

But when it comes to 2D HPs, the “plane” usually represents the abstract plane of the 2D HP slabs, e.g., the (h00) facets of Ruddlesden-Popper HPs and Dion-Jacobson HPs. As such, the in-plane direction denotes the direction parallel to the (h00) facets of 2D HPs, and the out-of-plane direction denotes the direction normal to the (h00) facets of 2D HPs, as illustrated in RI_Figure 2. This definition is widely adopted in the community. In the following, we provide several references to bolster this claim.

RI_Figure 2. Definitions of the “in-plane” and “out-of-plane” directions for 2D HPs.

RI_Figure 3. Panel a, panel e, and panel i of Figure 2 in the reference *ACS Energy Lett.* **7**, 984-987 (2022).^[3] x and y denote the space coordinate axes.

In a recent work (*ACS Energy Lett.* **7**, 984-987 (2022)) by Prof. Jinsong Huang,^[3] a well-known field expert, the authors gave the definitions of the “in-plane” and “out-of-plane” directions for 2D HPs (RI_Figure 3). In RI_Figure 3a, both the white arrow and the black arrow mark “in-plane” directions as they are parallel to the (h00) facets of the 2D HP. In RI_Figure 3b-3c, the black arrows still mark an “in-plane” direction, but the white arrows turn to represent an “out-of-plane” direction as they are normal to the (h00) facets of the 2D HP. When referring to the substrate (not presented in RI_Figure 3), however, all these directions should be called “in-plane” directions as they are parallel to the substrate.

RI_Figure 4. Figure 1c of the reference *Nature Nanotechnology* **17**, 45-52 (2022) with modification.^[4]

In another recent work (*Nature Nanotechnology* **17**, 45-52 (2022)) by Prof. Aditya D. Mohite,^[4] who is also a well-known field expert, the same definitions are introduced. As shown in RI_Figure 4. Clearly, the so-called “plane” refers to the (h00) crystallographic facets of 2D HPs (the blue line in the left panel and the blue rectangles in the right two panels). The direction normal to the (h00) facets is the “out-of-plane” direction (the left panel), and the direction parallel to the (h00) facets is the “in-plane direction” (the right top panel).

RI_Figure 5. Figure 1d of the reference *Nat. Commun.* **13**, **138** (2022).^[5]

RI_Figure 5 presents the same definitions in a recent work *Nat. Commun.* **13**, **138** (2022) by Prof. Kian Ping Loh (also a well-known expert) and coworkers.^[5] The $[Pb]_c$ tilt component (θ) perpendicular to the (h00) facets is called “out-of-plane tilt”, and

the $[\text{PbI}_6]$ tilt component parallel to the (h00) facets (γ) is called “in-plane tilt”.

Besides, there is a widely accepted opinion in this community that “in 2D HPs, the out-of-plane carrier transport is inferior to the in-plane carrier transport”.^[6-8] The “out-of-plane” carrier transport denotes that through the organic spacer layer, *i.e.*, vertical to the (h00) facets of Ruddlesden-Popper 2D HPs. The “in-plane” carrier transport denotes that within the HP skeleton. We do not need to figure out the meaning of this claim by referring to a substrate.

RI_Figure 6. Revised Figure 6 in the manuscript.

With these basic definitions in mind, the accuracy of the arrows in Figure 6 in the manuscript (*i.e.*, RI_Figure 6) and the associated descriptions regarding the “in-plane” and “out-of-plane” directions are rechecked as below: In the **a** panel, the direction parallel to the (h00) facets is denoted as the “in-plane” direction, and the direction parallel to (h00) facets is denoted as the “out-of-plane” direction. The same denotations apply to the **b** and **c** panels. The **d** panel describes the real phase distribution and orientation of typical mixed-dimensional 2D/3D HP solar cells, where the 2D HP fragments adopt a vertical orientation with the (h00) facets normal to the substrate. As such, the vertical carrier transport is parallel to the (h00) facets of 2D HPs and thus is denoted as an “in-plane” direction.

In order to make a clearer expression to avoid possible misleading, we have added the above definition in the revised manuscript (Page 16) as below. Besides, for the readers to better tell the in-plane direction and the out-of-plane direction in Figure 6, we have assigned arrows with different colors to indicate them (purple arrow for the in-plane direction and blue arrow for the out-of-plane direction).

“...From the perspective of space, the PCB effect is confirmed along the out-of-plane direction of 2D HPs in the above GIHP film (the “plane” here refers to the (100) facets of 2D HPs).^{[44]-[46]} But due to a vertical orientation of the 2D component in typical mixed-dimensional 2D/3D HP solar cells (Figure S19), carriers are expected to transport along the in-plane direction...”

(44) Shi, Z.; Ni, Z.; Huang, J. Direct observation of fast carriers transport along out-of-plane direction in a Dion–Jacobson layered perovskite. *ACS Energy Lett.* **7**, 984-987 (2022).

(45) Li, W.; Sidhik, S.; Traore, B.; Asadpour, R.; Hou, J.; Zhang, H.; Fehr, A.; Essman, J.; Wang, Y.; Hoffman, J. M.; Spanopoulos, I.; Crochet, J. J.; Tsai, E.; Strzalka, J.; Katan, C.; Alam, M. A.; Kanatzidis, M. G.; Even, J.; Blancon, J. C.; Mohite, A. D. Light-activated interlayer contraction in two-dimensional perovskites for high-efficiency solar cells. *Nat. Nanotechnol.* **17**, 45-52 (2022).

(46) Shao, Y.; Gao, W.; Yan, H.; Li, R.; Abdelwahab, I.; Chi, X.; Rogee, L.; Zhuang, L.; Fu, W.; Lau, S. P.; Yu, S. F.; Cai, Y.; Loh, K. P.; Leng, K. Unlocking surface octahedral tilt in two-dimensional Ruddlesden-Popper perovskites. *Nat. Commun.* **13**, 138 (2022).

We sincerely hope the above response addresses your concern and justifies the publication of this manuscript in *Nature Communications*. Any further comments are also highly welcomed. Thank you again for your valuable time and patience.

Reviewer #2 (Remarks to the Author):

The authors have tried to address the concerns of the reviewer. However, some concerns have not been fully addressed. These comments need to be addressed before publication. For example,

1. The in plane (organic layer parallel to the substrate) and out of plane (organic layer vertical to the substrate) directions of 2D perovskites are not fully understood.

Response: Dear reviewer, thank you very much for your time and comments. We noticed that your definitions of “in-plane” and “out-of-plane” directions were different from ours in the first round of peer review. We understand that your definition of “plane” is the substrate plane, therefore, the “in-plane” direction denotes a direction parallel to the substrate, and the “out-of-plane” direction denotes a direction normal to the substrate. As such, for typical mixed-dimensional 2D/3D HP solar cells with a vertical orientation of the 2D HP fragments, carriers migrate in the “out-of-plane” direction. Such definition, as far as we know, is more often adopted for 3D HP-based vertical devices (as shown in R2_Figure 1),^[1-2] because 3D HPs have a quasi-symmetric structure, and a direction could only be indicated by referring to the substrate.

R2_Figure 1. The definition of “in-plane” and “out-of-plane” directions in 3D HP films. X-ray diffraction in the (a) “out-of-plane” direction and (b) the “in-plane” direction. This figure is taken from the reference *Sci. Adv.* **3**, eaao5616 (2017) (Figure 1c and 1d).^[1]

But when it comes to 2D HPs, the “plane” is usually defined as the abstract plane defined by their crystallographic facets, e.g., the (h00) facets of Ruddlesden-Popper HPs and Dion-Jacobson HPs. Therefore, as illustrated in R2_Figure 2, the “in-plane”

direction denotes a direction parallel to the (h00) facets, and the “out-of-plane” direction denotes a direction normal to the (h00) facets. Following this definition, for typical mixed-dimensional 2D/3D HP solar cells with a vertical orientation of the 2D HP fragments, carriers migrate in the “in-plane” direction, as we described in the manuscript. With all due respect, you can see that this statement is exactly contrary to the above one, although they intend to express the same meaning.

R2_Figure 2. Definitions of the “in-plane” and “out-of-plane” directions in 2D HPs.

R2_Figure 3. Panel a, panel e, and panel i of Figure 2 in the reference *ACS Energy Lett.* **7**, 984-987 (2022).^[3] x and y denote the space coordinate axes.

The definition we adopted is widely accepted in the community. In the following, we provide several references to bolster this claim.

In a recent work (*ACS Energy Lett.* **7**, 984-987 (2022)) by Prof. Jinsong Huang,^[3] a well-known field expert, the authors gave the definitions of the “in-plane” and “out-of-plane” directions for 2D HPs (R2_Figure 3). In R2_Figure 3a, both the white arrow and the black arrow mark “in-plane” directions as they are parallel to the (h00) facets of the 2D HP. In R2_Figure 3b-3c, the black arrows still mark an “in-plane” direction, but the white arrows turn to represent an “out-of-plane” direction as they are normal to the (h00) facets of the 2D HP. When referring to the substrate (not presented in R2_Figure 3), however, all these directions should be called “in-plane” directions as they are parallel to the substrate.

R2_Figure 4. Figure 1c in the reference *Nature Nanotechnology* **17**, 45-52 (2022) with modification.^[4]

In another recent work (*Nature Nanotechnology* **17**, 45-52 (2022)) by Prof. Aditya D. Mohite,^[4] who is also a well-known field expert, the same definitions are introduced. As shown in R2_Figure 4. Clearly, the so-called “plane” refers to the (h00) crystallographic facets of 2D HPs (the blue line in the left panel and the blue rectangles in the right two panels). The direction normal to the (h00) facets is the “out-of-plane” direction (the left panel), and the direction parallel to the (h00) facets is the “in-plane direction” (the right top panel).

R2_Figure 5. Figure 1d in the reference *Nat. Commun.* **13**, **138** (2022).^[5]

R2_Figure 5 presents the same definitions in a recent work *Nat. Commun.* **13**, **138** (2022) by Prof. Kian Ping Loh (also a well-known expert) and coworkers.^[5] The $[Pb]_c$ tilt component (θ) perpendicular to the (h00) facets is called “out-of-plane tilt”, and

the $[\text{PbI}_6]$ tilt component parallel to the (h00) facets (γ) is called “in-plane tilt”.

Besides, there is a widely accepted opinion in this community that “in 2D HPs, the out-of-plane carrier transport is inferior to the in-plane carrier transport”.^[6-8] The “out-of-plane” carrier transport denotes that through the organic spacer layer, *i.e.*, vertical to the (h00) facets of Ruddlesden-Popper 2D HPs. The “in-plane” carrier transport denotes that within the HP skeleton. We do not need to figure out the meaning of this claim by referring to a substrate.

R2_Figure 6. Schematic illustration of the phase orientation of the mixed-dimensional 2D/3D HPs and the carrier transport (a) in the lateral ITO/GIHP film/ITO configuration and (b) in the vertical ITO/HP/Cu configuration.

With these basic definitions in mind, the charge carrier transport directions in our manuscript are reviewed as below. R2_Figure 6 presents the phase orientation and the carrier transport direction in the lateral ITO/GIHP film/ITO configuration and the vertical ITO/HP/Cu configuration. In the lateral ITO/GIHP film/ITO configuration (R2_Figure 6a), electrons transport through the (h00) facets, therefore, the transporting direction is marked as “out-of-plane”; In the vertical ITO/HP/Cu configuration (R2_Figure 6b), electrons transport parallel to the 2D HP planes, the transporting direction thus is marked as “in-plane”.

In order to make a clearer expression to avoid possible misleading, we have added the above definition in the revised manuscript (Page 16) as below:

“...From the perspective of space, the PCB effect is confirmed along the out-of-plane direction of 2D HPs in the above GIHP film (the “plane” here refers to the (100) facets of 2D HPs).^{[44]-[46]} But due to a vertical orientation of the 2D component in typical mixed-dimensional 2D/3D HP solar cells (Figure S19), carriers are expected to transport along the in-plane direction...”

(44) Shi, Z.; Ni, Z.; Huang, J. Direct observation of fast carriers transport along out-of-plane direction in a Dion–Jacobson layered perovskite. *ACS Energy Lett.* **7**, 984-987 (2022).

(45) Li, W.; Sidhik, S.; Traore, B.; Asadpour, R.; Hou, J.; Zhang, H.; Fehr, A.; Essman, J.; Wang, Y.; Hoffman, J. M.; Spanopoulos, I.; Crochet, J. J.; Tsai, E.; Strzalka, J.; Katan, C.; Alam, M. A.; Kanatzidis, M. G.; Even, J.; Blancon, J. C.; Mohite, A. D. Light-activated interlayer contraction in two-dimensional perovskites for high-efficiency solar cells. *Nat. Nanotechnol.* **17**, 45-52 (2022).

(46) Shao, Y.; Gao, W.; Yan, H.; Li, R.; Abdelwahab, I.; Chi, X.; Rogee, L.; Zhuang, L.; Fu, W.; Lau, S. P.; Yu, S. F.; Cai, Y.; Loh, K. P.; Leng, K. Unlocking surface octahedral tilt in two-dimensional Ruddlesden-Popper perovskites. *Nat. Commun.* **13**, 138 (2022).

2. The authors claim that the photoinduced carrier blocking (PCB) effect exists both in-plane and out plane direction in 2D perovskites. For mixed dimensional 2D/3D HP solar cells, the authors discussed in the manuscript as follows: “Although freely moving in darkness, electrons would be greatly hampered under illumination, and holes would be also deeply trapped.” While for the 2D-on-3D bilayer configuration, the authors don’t think the PCB effect play the roles.

Response: The term “PCB” is defined as an abbreviation of “photoinduced carrier blocking” in our manuscript, it is a phenomenon caused by the light-enhanced built-in potential of the 2D/3D HP interface. It is for sure that the light-enhanced built-in potential would also occur in the 2D-on-3D bilayer configuration the same as in the mixed-dimensional 2D/3D system. However, it would not lead to the PCB effect.

R2_Figure 7. The effect of light-enhanced built-in potential of the 2D/3D HP interface in (a) the mixed-dimensional 2D/3D HP solar cells and (b) the 2D-on-3D HP solar cells.

Such a difference arises from the different phase arrangements of the two systems: In the mixed-dimensional 2D/3D HP system, 2D HP fragments intersperse in the 3D HP matrix, thus giving rise to a local 3D/2D/3D arrangement (R2_Figure 7a). Upon illumination, the built-in potential of the 2D/3D HP interface increases, and electrons that otherwise could freely transport through the local 3D/2D/3D site would be inhibited by the 2D HP fragment, causing the so-called PCB effect. In a similar way, holes would be deeply trapped by the 2D HP fragment. This is the meaning of our statement you commented: "...Although freely moving in darkness, electrons would be greatly hampered under illumination, and holes would be also deeply trapped..."

In contrast, for the 2D-on-3D bilayer configuration, there is no such local 3D/2D/3D arrangement. The 2D HP layer caps the top surface of the 3D HP layer to serve as not only a passivating layer but also a carrier transporting layer. Carriers generated in the 3D HP layer would pass through the top 2D HP layer before being collected by the real charge transporting layer (R2_Figure 7b). As such, the light-enhanced built-in potential of the 2D/3D interface would, in turn, promote carrier harvesting as it

facilitates carrier separation to suppress the lossy recombination.

A PCB effect in the 2D-on-3D bilayer configuration could only occur when the carrier transporting layers are wrongly placed, e.g., the 2D HP layer plays a role similar to the hole transporting layer but is coated by an electron transporting layer (also a hole blocking layer). However, this is not likely to take place in practice, as researchers usually would carefully check the band alignment of the 2D-on-3D bilayer configuration before the solar cell fabrication.

To make a clearer expression, R2_Figure 7 is added to the revised manuscript as Figure S25, and the associated description above is added as Supplementary Note 7.

3. The authors claim that ITO/HP/Cu is also an electron-only configuration, which excludes the interference from bipolar transport. Why? Typically, electron transport layer (ETL) should be included in configuration, for example, ITO/ETL/HP/ETL/Cu.

Response: Yes, we agree that electron-transporting layers are usually required for an electron-only device, but they are not necessary.^[9-11] An illustrative example is shown in R2_Figure 8a,^[11] if the VBM (valence band minimum) of the semiconductor layer well aligns with the Fermi level of the metal electrodes, holes could freely pass through the metal/semiconductor interface, while electrons would be blocked by the Schottky barrier (approximately the bandgap of the semiconductor layer). As such, it will be a hole-only device. In a similar way, an electron-only device could also be obtained when the CBM (conduction band maximum) of the semiconductor layer aligns with the Fermi level of the metal electrode (R2_Figure 8b).

R2_Figure 8. (a) Schematic diagram of a hole-only device without hole-transporting layers. (b) Schematic diagram of an electron-only device without electron-transporting layers. This figure is taken from Ref.^[11]

In our work, the VBM positions of the mixed-dimensional 2D/3D HP film were determined by the depth-profile UPS measurement (Figure S6-S7, presented in the Supporting Information file), and the values are in the range of -6.24 eV ~ -6.35 eV relative to the vacuum level (R2_Figure 9, as stated in the manuscript, Page 6).

Considering the bandgaps of different HP phases (~ 2.17 eV for $n=2$; ~ 2 eV for $n=3$; 1.9 eV for $n=4$; ~ 1.6 eV for $n=\infty$) and the Fermi levels of ITO and Cu (-4.62 eV ~ -4.75 eV for ITO and -4.7 eV for Cu), it is not difficult to infer a significantly lower Schottky barrier for electrons than holes, especially considering that in mixed-dimensional 2D/3D HP films the 3D HP phase is dominant compared to the 2D ones, as illustrated in R2_Figure 10a.

R2_Figure 9. Depth-profile absolute VBM positions versus the vacuum level.

R2_Figure 10. (a) Schematic diagrams of the free transport of electrons and the inhibited hole transport in the ITO/mixed-dimensional HP/Cu configuration. The black region represents the 3D HP matrix, and the red regions represent the interspersing 2D HP phases. (b) The corresponding I-V characteristics obtained in darkness (the top panel) and under one-sun illumination (the bottom panel).

This claim is further supported by the I-V characteristics, which show an ohmic contact property both in darkness and under illumination (R2_Figure 10b). It is

impossible to realize an ohmic contact for both electrons and holes in such a metal/semiconductor/metal configuration, which underpins the electron-only nature.

Overall, we sincerely hope the above response addresses your concerns and justifies the publication of this manuscript in *Nature Communications*. Any further comments are also highly welcomed. Thank you again for your valuable time and patience.

Reviewer #3 (Remarks to the Author):

The authors have already addressed most of the issues from the reviewers, and the paper could be accepted with this version.

Response: Dear reviewer, thank you very much for your time and comments, which helped a lot in improving this work. We sincerely hope the further revisions made according to the comments of Reviewer #1 and Reviewer #2 would justify the publication of our work in *Nature Communications*.

Reference

- (1) Zhao, J.; Deng, Y.; Wei, H.; Zheng, X.; Yu, Z.; Shao, Y.; Shield, J. E.; Huang, J. Strained hybrid perovskite thin films and their impact on the intrinsic stability of perovskite solar cells. *Sci. Adv.* **3**, eaao5616 (2017).
- (2) Liu, D.; Luo, D.; Iqbal, A. N.; Orr, K. W. P.; Doherty, T. A. S.; Lu, Z. H.; Stranks, S. D.; Zhang, W. Strain analysis and engineering in halide perovskite photovoltaics. *Nat. Mater.* **20**, 1337-1346 (2021).
- (3) Shi, Z.; Ni, Z.; Huang, J. Direct observation of fast carriers transport along out-of-plane direction in a Dion–Jacobson layered perovskite. *ACS Energy Lett.* **7**, 984-987 (2022).
- (4) Li, W.; Sidhik, S.; Traore, B.; Asadpour, R.; Hou, J.; Zhang, H.; Fehr, A.; Essman, J.; Wang, Y.; Hoffman, J. M.; Spanopoulos, I.; Crochet, J. J.; Tsai, E.; Strzalka, J.; Katan, C.; Alam, M. A.; Kanatzidis, M. G.; Even, J.; Blancon, J.-C.; Mohite, A. D. Light-activated interlayer contraction in two-dimensional perovskites for high-efficiency solar cells. *Nat. Nanotechnol.* **17**, 45-52 (2022).
- (5) Shao, Y.; Gao, W.; Yan, H.; Li, R.; Abdelwahab, I.; Chi, X.; Rogee, L.; Zhuang, L.; Fu, W.; Lau, S. P.; Yu, S. F.; Cai, Y.; Loh, K. P.; Leng, K. Unlocking surface octahedral tilt in two-dimensional Ruddlesden-Popper perovskites. *Nat. Commun.* **13**, 138 (2022).
- (6) Fang, Z.; Hou, X.; Zheng, Y.; Yang, Z.; Chou, K. C.; Shao, G.; Shang, M.; Yang, W.; Wu, T. First-principles optimization of out-of-plane charge transport in Dion-Jacobson CsPbI₃ perovskites with π -conjugated aromatic spacers. *Adv. Funct. Mater.* **31**, 2102330 (2021).
- (7) Passarelli, J. V.; Fairfield, D. J.; Sather, N. A.; Hendricks, M. P.; Sai, H.; Stern, C. L.; Stupp, S. I. Enhanced out-of-plane conductivity and photovoltaic performance in $n = 1$ layered perovskites through organic cation design. *J. Am. Chem. Soc.* **140**, 7313-7323 (2018).
- (8) Kober-Czerny, M.; Motti, S. G.; Holzhey, P.; Wenger, B.; Lim, J.; Herz, L. M.; Snaith, H. J. Excellent long-range charge-carrier mobility in 2D perovskites. *Adv. Funct. Mater.* **n/a**, 2203064 (2022).
- (9) Sajedi Alvar, M.; Blom, P. W. M.; Wetzelaer, G.-J. A. H. Space-charge-limited electron and hole currents in hybrid organic-inorganic perovskites. *Nat. Commun.* **11**, 4023 (2020).

- (10) Röhr, J. A. Direct determination of built-in voltages in asymmetric single-carrier devices. *Phys. Rev. Appl.* **11**, 054079 (2019).
- (11) Ferreira, A. d. C. In *Perylene diimide acceptors: fabrication and characterization of electron-only, hole-only devices and solar cells*, 2015.

REVIEWERS' COMMENTS

Reviewer #1 (Remarks to the Author):

The authors provide detailed experimental supplements and discussions of previous issues that address concerns about the previous manuscript, therefore I recommend the revised manuscript for publication.

Reviewer #2 (Remarks to the Author):

The authors have addressed all the concerns'of the reviewers. I think the manuscript should be accepted for publication in Nature Communication.

Reviewer #1 (Remarks to the Author):

The authors provide detailed experimental supplements and discussions of previous issues that address concerns about the previous manuscript, therefore I recommend the revised manuscript for publication.

Response: Dear reviewer, thank you very much for your time and comments, which helped a lot in improving this work. We sincerely hope the renewed insights provided by our work would contribute to improving the performance of mixed-dimensional 2D/3D perovskite solar cells.

Reviewer #2 (Remarks to the Author):

The authors have addressed all the concerns'of the reviewers. I think the manuscript should be accepted for publication in Nature Communication.

Response: Dear reviewer, thank you very much for your time and comments, which helped a lot in improving this work. We sincerely hope the renewed insights provided by our work would contribute to improving the performance of mixed-dimensional 2D/3D perovskite solar cells.